# Analysis of micro-seismicity in sea ice with deep learning and Bayesian inference: application to high-resolution thickness monitoring

Ludovic Moreau[1], Léonard Seydoux[1], Jerome Weiss[1], and Michel Campillo[1]

[1]Institut des Sciences de la Terre, Université Grenoble Alpes, Grenoble, France

**Correspondence:** Ludovic Moreau (ludovic.moreau@univ-grenoble-alpes.fr)

**Abstract.** In the perspective of upcoming seasonally ice-free Arctic, understanding the dynamics of sea ice in the changing climate is a major challenge in oceanography and climatology. In particular, the new generation of sea ice models will require fine parameterization of sea ice thickness and rheology. With the rapidly evolving state of sea ice, achieving better accuracy, as well as finer temporal and spatial resolutions of its thickness will set new monitoring standards, with major scientific and geopolitical implications. Recent studies have shown the potential of passive seismology to monitor the thickness, density and elastic properties of sea ice with significantly reduced logistical constraints. For example, human intervention is no longer required, except to install and uninstall the geophones. Building up on this approach, we introduce a methodology for estimating sea ice thickness with high spatial and temporal resolutions from the analysis of icequakes waveforms. This methodology is based on a deep convolutional neural network for automatic clustering of the ambient seismicity recorded on sea ice, combined with a Bayesian inversion of the clustered waveforms. By applying this approach to seismic data recorded in March 2019 on fast ice in the Van Mijen fjord (Svalbard), we observe the spatial clustering of icequakes sources along the shore line of the fjord. The ice thickness is shown to follow an increasing trend that is consistent with the evolution of temperatures during the four weeks of data recording. Comparing the energy of the icequakes with that of artificial seismic sources, we were able to derive a power law of icequake energy, and to relate this energy to the size of the cracks that generate the icequakes.

## 1 Introduction

With the rapidly evolving climate in polar regions, collecting field data is key for anticipating the major upcoming changes related to global warming. In particular, sea ice is an essential element of polar regions because of the role it plays in phytoplankton production (Mayot et al., 2020), and in several atmosphere-ice-oceans interactions. In the Arctic, the extent of sea ice in summer undergoes an important negative trend of about 12.6% per decade, according to the National Snow and Ice Data Center. In the Antarctic, Parkinson (2019) observed a weak positive trend of 1.5% per decade. However, this positive trend should be mitigated by the outstanding and unprecedented decline in 2015-2017, which shows how vulnerable Antarctic sea ice is to both ocean warming and changes in large-scale atmospheric winds (Eayrs et al., 2019). This emphasizes the need for progress in research addressing altogether the nature of these changes, their pace, and also their impact at the global scale. In this matter, a finer and more accurate description of the dynamics and thermodynamic processes of sea ice is needed.

Given the challenging logistics for accessing polar regions, most of the knowledge about sea ice extent and concentration comes from remote sensing and in particular microwave-radar imagery, which provides altimetric information, which can be converted to ice thickness. Sea ice thickness is an important parameter, for many reasons. For example, thick ice filters light more than thin ice, hence thickness influences phytoplankton production (Ardyna et al., 2014). Thicker ice is also more resilient to external forcing such as swell or wind forcing (Dumont, 2022). Hence in the research effort for monitoring the state of sea ice, much focus is given to improving the spatial resolution and accuracy of thickness estimations. Remote sensing methods rely on a conversion between the freeboard measurement into an ice thickness estimation. These methods suffer from a lack of in situ measurements to calibrate the estimations, which can have several sources of errors, such as the presence of the snow cover, as well as uncertainties on the freeboard, on the densities of ice and snow, etc. (Garnier et al., 2021).

Seismic methods have been shown to be good candidates for evaluating sea ice properties at the local scale, with very good accuracy and spatial resolution. The first seismic experiments on sea ice date back to the late 1950s, where the elastic constants and the thickness of sea ice were estimated from the velocity of the seismic waves travelling in the ice cover (Crary, 1954; Anderson, 1958; Hunkins, 1960). With the emergence of digital signal processing, methods based on Fourier analysis were made possible, allowing more accurate inversions of the signals to recover both the ice thickness and its elastic properties (Yang and Giellis, 1994; Stein et al., 1998). However, collecting seismic data on sea ice has long remained too challenging for applications to sea ice monitoring. With the miniaturisation of electronic components, the rapid progress in terms of battery life, and the era of seismic noise interferometry (Shapiro and Campillo, 2004; Sabra et al., 2005), it has become possible to collect data without the need of active, human-controlled sources, and then to process them remotely (Marsan et al., 2019). Therefore, over the past decade, there has been a renewed interest in seismic methods as a complementary means of monitoring the thickness, density and elastic properties of sea ice (Marsan et al., 2012; Moreau et al., 2020a, b; Romeyn et al., 2021; Serripierri et al., 2022).

The missing link between data acquisition and long-term sea ice monitoring is the ability to extract, in the continuous recordings, the useful parts of the seismic waveforms from the background noise, for automatic estimations of the sea ice properties. In this paper we combine a deep learning method for automatic clustering of the waveforms (Seydoux et al., 2020) recorded on sea ice with Bayesian inference to locate the position of thousands of icequakes while simultaneously evaluating the ice thickness. We demonstrate the possibility of generating maps of sea ice thickness and microseismic activity, with a temporal resolution that is directly linked to the amount of icequakes recorded. In the specific configuration at the fjord, icequakes occurrences are driven essentially by tide. On drifting ice, icequakes are generated by other mechanisms such as swell or ice motion, and many icequakes are also present in the ambient seismic field (Moreau et al., 2020b). With hundreds of icequakes recorded everyday, a daily temporal resolution can be achieved. We also use the energy information to calculate the scaling law of icequakes in terms of their released energy.

## 2 Instruments and methods

### 2.1 Seismic array

In a thin structure such as sea ice, the seismic wavefield is multiply reflected at the upper and lower interfaces of the ice. These reflections interfere constructively to produce guided modes that propagate in the elongated direction of sea ice. When the product of the ice thickness by the frequency is larger than 1000 Hz·m, the fundamental modes co-exist with higher order modes in the wavefield (Moreau et al., 2017). Here, we restrict our analyses to frequency × thickness values where only the fundamental modes are propagating. We use the terminology introduced in Moreau et al. (2020a) for referring to these three modes:

- the quasi-Scholte mode ($QS$), also known in the low-frequency regime as the flexural gravity wave;

- the fundamental quasi-symmetric mode ($QS_0$), also known in the low-frequency regime as the longitudinal wave;

- the fundamental shear-horizontal mode ($SH_0$), which produces shear-horizontal motion.

The $QS$ mode is highly dispersive at low frequencies, hence seismic signals recorded in sea ice away from the source are distorted. It is noteworthy that the $SH_0$ mode is not dispersive and that the $QS_0$ becomes dispersive only at higher frequencies. An important property of guided wave propagation is that given a set of ice mechanical properties, there is a direct relationship between the dispersion of the waveforms, the waveguide thickness and the source-receiver distance. By recording the seismic wavefield in sea ice, it is therefore possible to recover the structural information of the ice. In the frequency regime of interest here, only the $QS$ mode is dispersive, and it has most of its energy on the vertical component of the displacement, while the $QS_0$ and $SH_0$ modes are not dispersive and have energy mainly on the horizontal components of the displacement.

To record the seismic wavefield in sea ice, an experiment was conducted on fast ice in Svalbard (Norway), in a specific part of the Van Mijen Fjord called Vallunden (Figure 1a). This part of the fjord is surrounded by moraines, and can therefore be regarded as a "lake connected to the fjord", with a depth of about 10 m (Marchenko and Morozov, 2013). A dense array of 247 geophones was deployed and left to record seismic noise for one month. For a detailed description of the dense array, we refer the reader to Moreau et al. (2020a). In the present study, however, we use only five geophones of the dense array, as shown in Figure 1b as stations $S_i$ ($i = 1, 2, ..., 5$). Fast ice in this place of the fjord was continuous, with cracks located for the most part along the shore line, which indicates that they are likely tide cracks.

The data in this paper were recorded using FairFieldNodal Zland three-component geophones. These all-in-one sensors have an internal battery, a built-in GPS, and flash memory to store the data, allowing several weeks of autonomous and continuous recording, without the need of external cables. They have a cylindrical geometry of about 17 cm in height and 12 cm in diameter, and are mounted on a detachable spike. They were installed directly in the ice without their spike. To maximize the coupling, a milling tool was specifically designed to drill the ice at the diameter of the nodes. The snow was removed prior to drilling holes, and geophones were installed in the holes at about half their height. We covered them back with snow to

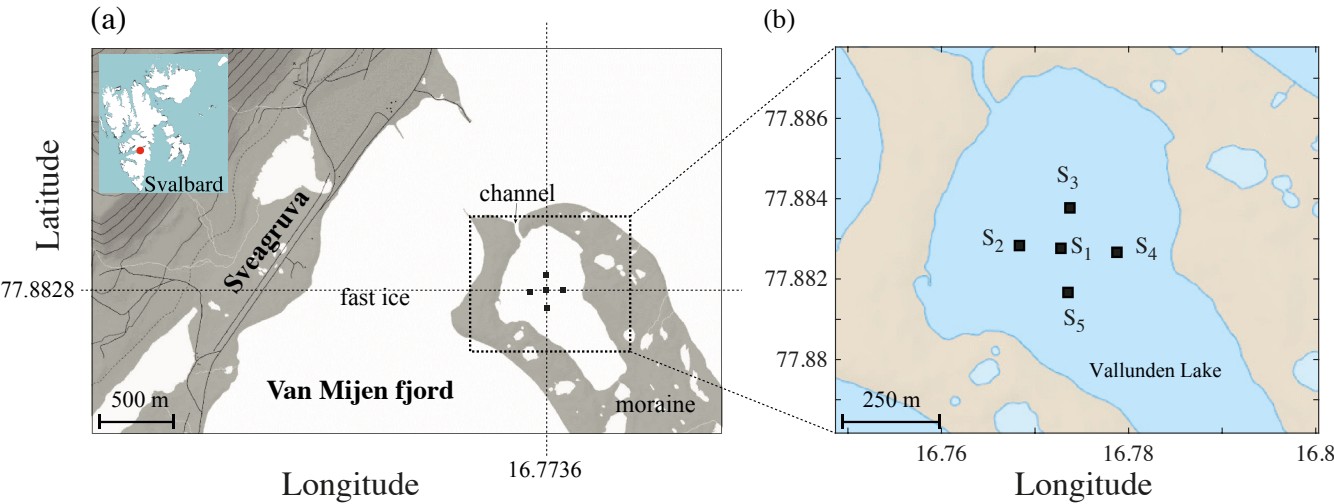

**Figure 1.** (a) Map of the area where the seismic array was installed (black squares) on Lake Vallunden, Svalbard. It is naturally bounded by moraines and connected to Van Mijen Fjord by a small channel. The facilities of Sveagruva are located about 1.5 km west of the deployment. (b) Zoom on the five seismic stations, $S_i$ ($i = 1, 2, ..., 5$), used to record the ambient seismic field. Arrows indicate the positions of ice drillings (ID). ID1 was performed on March 1, giving a thickness of 62 cm, ID2 and ID3 were performed on March 26, giving thicknesses of 70 and 73 cm.

insulate them in view of preserving their battery life. At the time of the deployment, the internal temperature of several nodes was measured, before and after covering them with snow, showing an increase from -21 to -16 °C.

FairFieldNodal Zland three-component geophones have a flat frequency response down to their eigenfrequency of 5 Hz. We recorded the data with a sampling frequency of 1000 Hz (Moreau and RESIF, 2019), but in order to reduce the computational cost of the present study, data were downsampled at 250 Hz. Conversion of the raw data into a miniseed format was obtained using the Fairfield software. Instrument response deconvolution is not necessary for our methodology and was therefore not applied. Data are expressed in mV, but could be converted to velocity by dividing by the proportionality factor 89 V/m/s, and further converted to displacement by integration with respect to time.

## 2.2 Automatic clustering of the waveforms

In Moreau et al. (2020b), we introduced an approach based on a Bayesian inversion of the icequakes waveform to recover the ice thickness while simultaneously relocating the source position, after the Young's modulus and Poisson's ratio of the ice were estimated from noise interferometry. This method was validated on a few icequakes recorded in fast ice and in pack ice. In this paper we are using this approach to conduct a systematic analysis of the thousands of icequakes recorded at Vallunden during the one-month experiment.

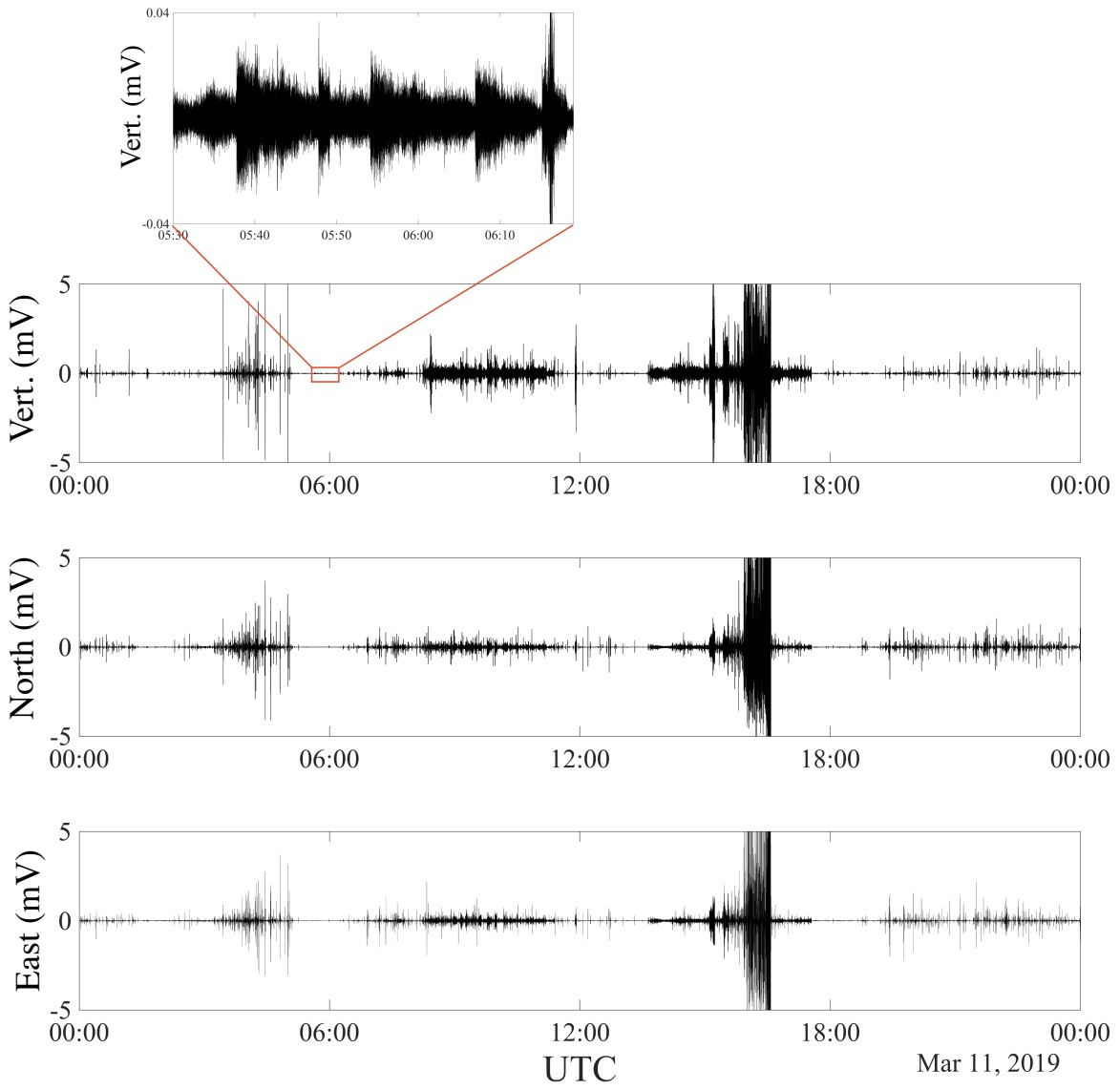

**Figure 2.** The three waveforms components of 24 hours of ambient seismic field recorded on fast ice in Svalbard on 11 March 2019. The wavefield is very rich and composed of icequakes with different orders of energy magnitude, transients (see the zoomed area on the vertical channel), and anthropogenic activities (between 8 AM and 5 PM).

Figure 2 shows 24 hours of recording on 11 March 2019, at station $S_1$ (see Figure 1b). This typical recording exhibits various types of signals, including thousands of icequakes with several orders of magnitude of energy, long-lasting transients, and background noise. Some waveforms are also related to anthropogenic activities. Therefore, the first processing step consists of extracting all icequakes from the recordings. Template matching is a common processing method for detecting similar waveforms in continuous recordings. However, although icequakes may look similar, their propagation in sea ice is accompanied by a strong dispersion of the quasi-Scholte mode. Hence, each combination of ice thickness and propagation distance results in a different waveform. For this reason, we use the Scatseisnet algorithm, introduced by Seydoux et al. (2020), and we apply this clustering algorithm to the three components of the displacements recorded at $S_1$.

Scatseisnet is a deep-learning inspired algorithm that automatically clusters segments of seismic data in continuous seismic records at a unique station. It combines a deep scattering network (Andén and Mallat, 2014) to transform the seismic waveforms into a relevant data space to identify relevant features suitable for clustering. The most relevant features are then extracted from the output of the deep scattering representation with an independent component analysis (Comon, 1992). A summary of these processing steps is given in Appendix B.

This strategy is applied on the segmented seismic time series, with a fixed window size of a few times the duration of the events of interest (see, e.g., Steinmann et al. (2022) for more information). For every signal segment, we obtain 20 real-valued features out of the independent component analysis. We finally use a hierarchical clustering approach to identify clusters of signal segments. By adjusting the distance threshold of the dendrogram (ward distance), we control the final number of clusters. A smaller distance implies a larger number of clusters. In the present case, apart from the several minutes-long transients waveforms, the typical duration of the waveforms in our recording is of a few seconds. Since this study does not focus on the transient signals, we use a 40s-long sliding window with a 20s overlap. We represent the hierarchical clustering output in a form of a dendrogram as in Figure 3a, for data collected at station $S_1$. This clustering would not change if applied to another stations, which is one of the advantages of using a deep scattering network. With the threshold distance indicated in the figure, we identify 6 families of signals, represented with different colors.

The family with clusters referenced 0 to 7 (figure 3a) cumulates about 30% of the dataset (figure 3b). Figure 4a shows 10 waveforms randomly sampled from cluster 0. We see that this cluster contains clean icequakes with very good SNR. So do the other 6 clusters of the first family. The icequakes have calendar occurrences every day of the deployment, but are more frequent between February 27th and March 13th, and then between March 21st and March 25th (figure 3c). It is noteworthy that this is consistent with the tides chart shown in figure 3f, and also with the fact that spring tides occurred on March 6 and March 21, while a neap tide occurred on March 13. Icequakes occur at all time of the day with the same temporal distribution, except around 9 AM where occurrences are slightly decreased (figure 3d). The decreased occurrence rates in the latter half of March could be due to the thickening of the ice ($\sim 25\%$ thickness increase).

Figure 3e indicates that the signals have a frequency content ranging between 1 and 50 Hz. To be more specific, on the vertical channel, dominated by the $QS$ mode, the amplitude of the spectrum of icequake waveforms remains (on average) over -30 dB between 1 and 35 Hz, with a peak value around 8 Hz. On the horizontal channels, where the $QS_0$ and $SH_0$ modes are dominant, the spectrum remains over -30 dB up to 50 Hz.

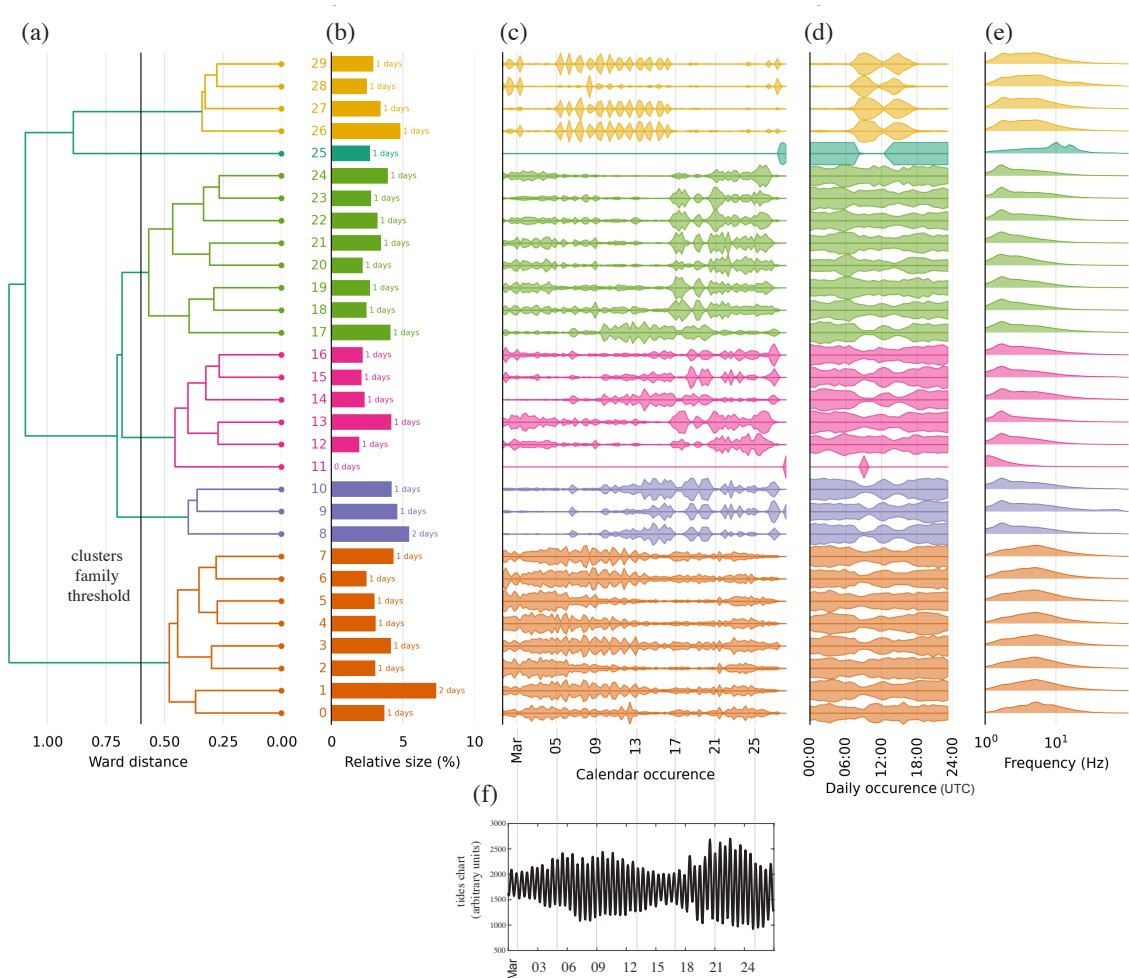

**Figure 3.** (a) Hierarchical representation of the ambient seismic field after the Scatseisnet algorithm was applied to the 27 days of continuous recording at station $S_1$. 6 families (shown as different colors) and 30 clusters are categorized. For each sub-family: (b) cumulated duration of the waveforms, (c) calendar occurrence, (d) time of day of occurrences, and (e) spectrum of the waveforms. (f) Tides chart at the Ny-Ålesund station ($\sim 160$ km away from Vallunden).

Although some icequakes may be the consequence of thermomechanical forcing (Olinger et al., 2019), it is likely that the majority are tidal icequakes. The temperature log can be extracted from the Sveagruva weather station SN99760 located 2 km west of the place of experiment. Data for this station are available at: https://seklima.met.no/hours/air_temperature/custom_period/SN99760/en/. These temperatures are shown in figure 5. The absence of a periodic pattern in temperature variations suggests that tides have more effect on the generation of icequakes than changes in temperature. The majority of icequakes occurs with a period of 24 hours (figure 6). This periodicity can also be seen in figure 3c, especially between March 1st and March 15th. One would expect semidiurnal tide (10-20 cm) to reflect in the periodicity of the icequakes, but the specific geometry of the moraines around

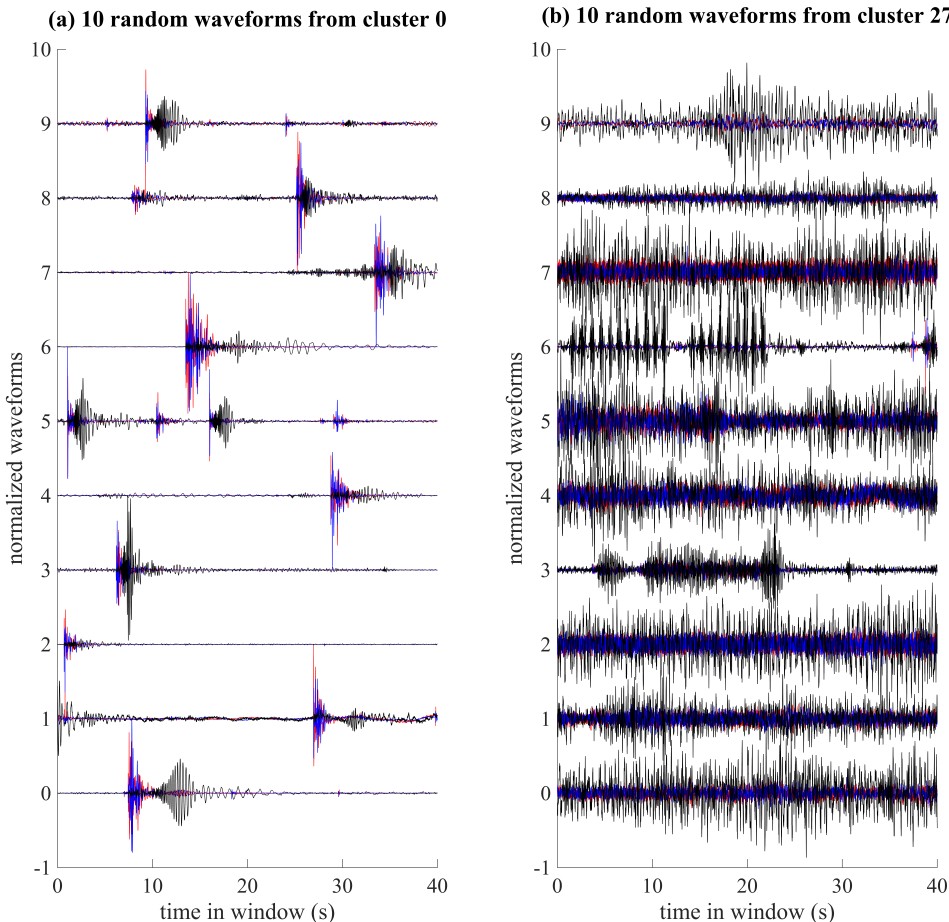

**Figure 4.** (a) Random waveforms extracted from cluster 0, which contains mainly clean icequakes. (b) Random waveforms extracted from cluster 27, which contains mainly signals from anthropogenic activities. Black: vertical component, blue: axial component, red: transverse component of displacement.

the experiment, together with the small channel that connects it to the fjord, generates some nonlinear effects that causes the tide in Vallunden to be asymmetric (Marchenko and Morozov, 2013). This could explain why occurrences are dominated by a period of 24h instead of 12h.

The family with clusters referenced 26 to 29 cumulates about 13% of the dataset. Figure 4b shows 10 waveforms randomly sampled from cluster 27. The waveforms are more complex and include many impulsive events, some noise, repeating events, and events that last up to 15-20 seconds. The waveforms in this family occur mostly during three sequences. The first sequence is between February 27th and March 2nd, when we went in the field to deploy the geophones and performed field experiments, including impulsive sources, sweep sources, snowmobile driving etc. The second sequence is between March

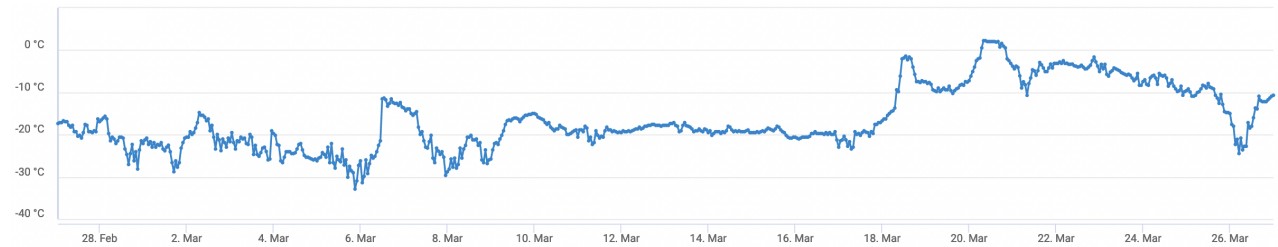

**Figure 5.** Air temperature recorded during deployment at the Sveagruva weather station (SN99760) located about 2 km west of the experiment. Temperatures remained between -30$^o$ and -10$^o$ until March 18th and then remained essentially between -10$^o$ and -0$^o$ until the end of deployment, which is consistent with the dynamic of ice thickness shown in figure 7, which exhibits a sharp increase between February 27th and March 16th, and then stabilizes.

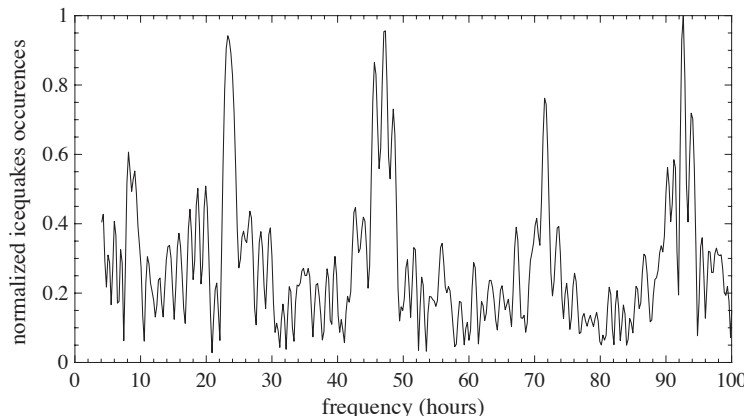

**Figure 6.** Frequency of icequakes occurrences, obtained by discrete Fourier transform of the dates of all icequakes in clusters 0-7, and normalized with the maximum amplitude in the resulting spectrum. The majority of icequakes occurs with a period of 24 hours, and are therefore most likely linked to tidal forcing.

5$^{th}$ and March 16$^{th}$, when a team of researchers and students were conducting field experiments about 150 m northeast of station $S_3$ (Marchenko et al., 2021). The third sequence is between March 25$^{th}$ and March 27$^{th}$, when we went back for some more field experiments before removing the geophones. For a list of exact coordinated universal time of the impulsive and sweep sources in sequences 1 and 3, please refer to Table A1 in Moreau et al. (2020a). All sequences occur between 8 AM and 6 PM (UTC), with a quieter time around 12:00. Hence we conclude that this family contains waveforms associated with anthropogenic activities.

The clusters of the other families contain either icequakes with low signal-to-noise ratio (SNR) in clusters 17-24, or waveforms that correspond to noise or that do not exhibit obvious correlation with surrounding activities. In the following, cluster 27 will be used for i) identifying events that correspond to the artificial impulsive sources reported in Moreau et al. (2020a)

160 and ii) calculating the associated source energy. Then, the energy of these artificial sources will be used to calibrate the energy of the icequakes extracted from clusters 0-7.

## 3 Results

### 3.1 Icequakes inversion

In this section, the methodology introduced in Moreau et al. (2020b) is applied to all waveforms in cluster 0 to 7, and also
to those in cluster 27. This represents a total of 5350 icequakes to invert for ice thickness, source coordinates, and source activation time. The other clusters were not analyzed further, for two reasons: first, because the waveforms in the other clusters have lower SNR, and second, for computational time economy. Inversion is based on the Markov Chain Monte Carlo (MCMC) algorithm, which requires tens of thousands of iterations for proper sampling of the parameters space. For this paper to be self-consistent, we briefly recall the inversion method, and the reader is invited to refer to Moreau et al. (2020b) for the practical
details of the implementation. The method consists of the following steps:

1. given a set of parameters for source position around the array (latitude and longitude), source activation time, and ice thickness, generate the synthetic waveforms of the $QS$ mode at the geophones. Synthetic waveforms are generated based on a Ricker wavelet that is propagated in the ice using the analytical, low-frequency asymptotic model by Stein et al. (1998), with the following ice mechanical properties: Young's modulus = 3.8 GPa, Poisson's ratio = 0.28, and density =
910 kg/m$^3$ (Serripierri et al., 2022). This model cannot account for the finite water depth of about 10 m, like, for example, the model by Romeyn et al. (2021) can, but by comparing both models, we have checked that this has negligible effect at the frequencies of interest.

2. replace the amplitude of the synthetic signals spectrum with that of the signals recorded at the geophones. This is meant to account for source mechanism in a simple way.

3. compute the time vs frequency spectra of the $QS$ mode in the synthetic and recorded waveforms, and calculate the cost function, defined as the $L2$ norm between these spectra

4. iterate with a MCMC scheme.

It is noteworthy that, although we make use of the three displacement components for clustering the waveforms with the Scatseisnet algorithm, in the inversion process we are only using the vertical displacement, where the $QS$ mode is measured.
The other two modes, measured on the horizontal displacement, are not sensitive to the ice thickness: the $SH_0$ mode is not dispersive (regardless the frequency), and the $QS_0$ is not dispersive at the considered frequency $\times$ thickness values. These modes are, however, sensitive to the density and elastic properties of the ice. It would therefore be possible to invert all parameters simultaneously by including the three modes in the cost function, but the waveforms of the $QS_0$ and $SH_0$ modes are not always clearly separated in time. Hence making the inversion process automatic requires a different inversion strategy,
such as full waveform inversion, which is much more computationally expensive.

Given the field conditions at the deployment site, the parameters space for the MCMC algorithm to explore is such that:

- the position of sources is within a distance of 1 km around station $S_1$,

- ice-thickness is comprised between 0.2 m and 1 m,

- the source activation time is within a 12-seconds window around the icequake recording time, to account for propagation time between the sources and the geophones.

Each inversion provides a probability density function for the parameters. After all inversions were performed, a quality check was applied to keep only those for which the standard deviation of the source position is less than 20 m, and that of the ice thickness is less than 2 cm, resulting in 1790 selected inversions. This does not mean that the non-selected inversions cannot be exploited, but we wanted to keep the best possible inversions, while retaining a sufficient amount of data for statistical analyses.

Figure 7a shows the map of the inversions that meet the quality threshold. One can see that sources are located essentially along the shore line, where most of the stress is concentrated. This is consistent with previous reports on the dynamics of tidal cracks. See for example the observation in the Van Mijen fjord by Caline and Barrault (2008).

The artificial impulsive sources near stations $S_3$ and $S_5$ are indicated with black arrows. These belong to cluster 27, associated with anthropogenic activities. These sources were generated by jumping onto the ice. Another set of artificial sources appears inside the area marked with a black square. These sources were realized during the above-mentioned field experiments that took place between March 5[th] and March 16[th]. Part of these experiments consisted of a floating ice block collisions with consolidated ice. The collisions were realized after Marchenko et al. (2021) cut a 5-meters large × 10-meters long × 0.8-meter high floe from the consolidated fast ice in the fjord. The floe with a mass of 39 tons, was then pulled with ropes to enter in collision with the surrounding fast ice. We checked with the authors of Marchenko et al. (2021) that the time of these events, which was extracted with the Scatseisnet algorithm, match the time of the collisions experiments. Interestingly, these events belong to cluster 0, which means that the Scatseisnet algorithm clusters the collisions in the family of icequakes instead of clustering them in the family of anthropogenic activities (cluster 27). The waveforms in clusters associated with anthropogenic activities have most of their energy on the vertical displacement component (figure 4b), while those associated with icequakes have more energy on the horizontal displacement components (figure 4a). Hence we deduce that the recorded icequakes are generated by source mechanisms dominated by traction/compression motion similar to the collisions source mechanisms, which is quite different from dislocation mechanisms encountered in fault zone for earthquakes. If these icequakes had similar dislocation mechanisms, an anti-symmetric motion (with respect to he middle-plane of the ice layer) would be generated and the energy would mainly go to the vertical displacement component.

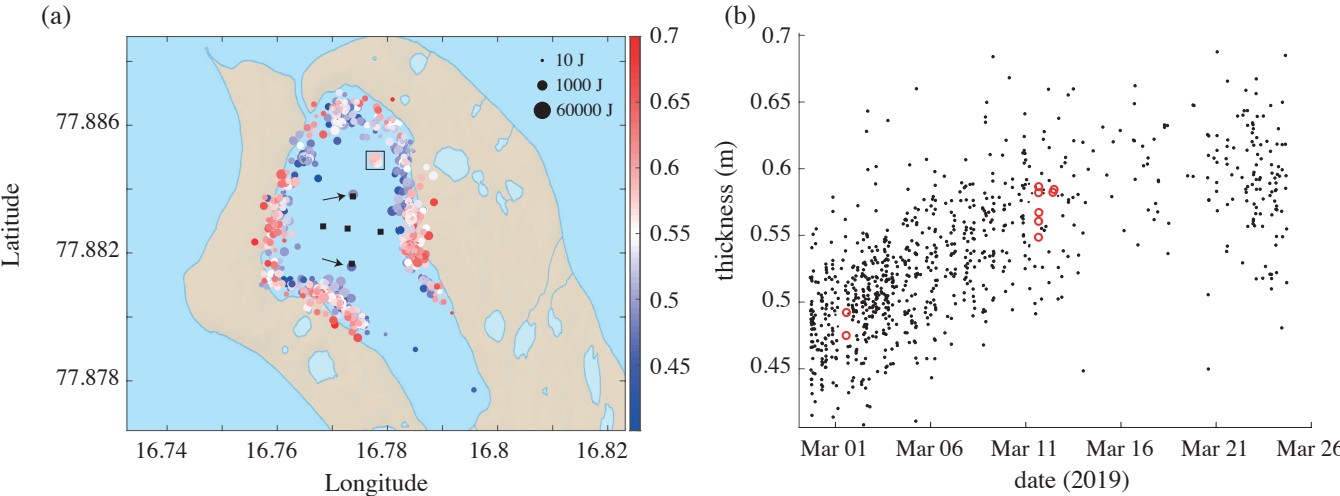

**Figure 7.** (a) Map of the inverted icequakes position. The colorbar indicates the corresponding ice thickness (with bounds on the thinnest and thickest values found). This thickness represents an average value along the paths between the icequake source and the 5 geophones. The size of the circles is proportional to the icequakes energy. The black arrows indicate the position of the vertical impulsive sources, and the black square that of the horizontal impulsive source. (b) Ice thickness versus date, obtained from icequakes inversions (black dots), and from waveforms with anthropogenic sources (red circles). The squares indicate values found by drilling the ice. We note a sharp increase in thickness during the first 2 weeks of deployment, followed by a stabilization during the remaining days, which is consistent with the raise of temperatures shown in figure 5

Figure 7a also shows, in color, the range of thicknesses associated with the 1790 selected inversions. Different thicknesses appear from identical positions, indicating that ice thickness has increased between the beginning and the end of the deployment. This is more visible in Figure 7b, where the inverted thicknesses are represented versus time. On average, the thickness increased by about 15 cm, which is consistent with the increase reported in Serripierri et al. (2022), but dispersion is more significant. This is because in the present paper, ice thickness is evaluated from all directions along the shore line, and covers a much larger range of distances from the stations (from 5 m to 1000 m) than in Serripierri et al. (2022), where it is evaluated along two lines of receivers oriented NS and EW, both lines having a length of 50 m. It is noteworthy that thickness estimates using only icequakes that are originating from a same region and at a similar date have a significantly reduced dispersion that is consistent with the standard deviation of each individual inversion, as shown in figure 8.

The ice thickness increase was also confirmed by ice drillings on March 1 and March 25. There is a sharp increase in thickness during the first two weeks, followed by a stabilization during the remaining days. This is consistent with the temperature shown in figure 5, which shows variations between -15 and -30 °C in the first half of March, while temperatures raised between -10 and 0 °C in the second half.

It is noteworthy that figure 7a represents all the recorded icequakes during the 27 days of geophones deployment. However, given the amount of icequakes recorded everyday, it could be possible to generate a similar map for each day, hence achieving near real-time maps of sea ice thickness evolution.

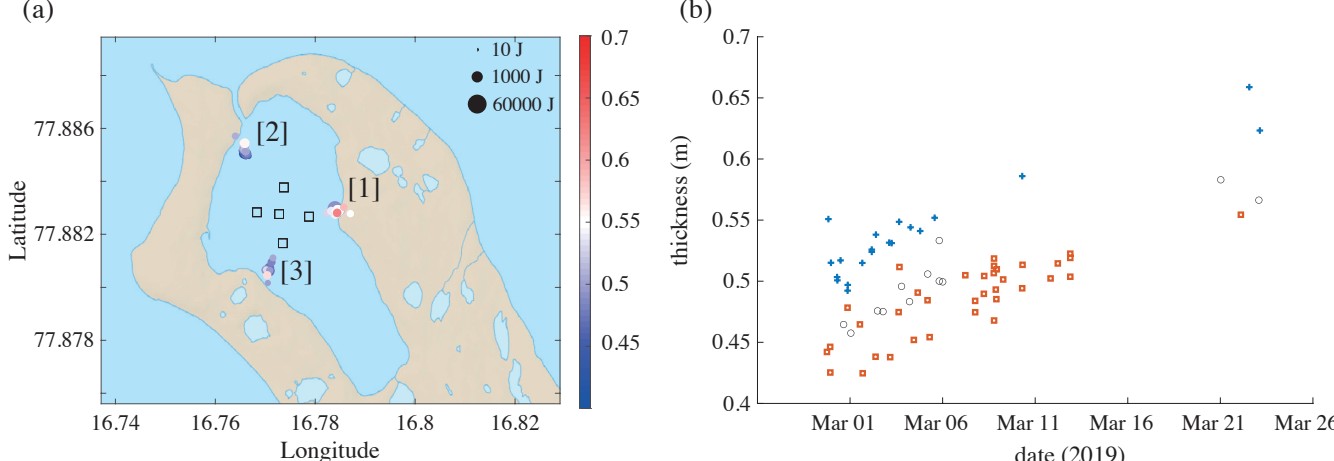

**Figure 8.** a) Same as figure 7a, restricted to icequakes originating only from directions marked as [1], [2] and [3]. b) same as figure 7b, for the icequakes in the three groups shown in 8a: + are for group [1], o are for group [2] and □ are for group [3]. Inversions originating from a same region and at a similar date have a range of thicknesses that is consistent with the standard deviation of each individual inversion (i.e. 2.5 cm). When comparing all directions, however, the range of thicknesses is of the order of 20 cm.

### 3.2 Energy of the artificial sources

Estimating the energy of the icequakes requires information about the decay of amplitude between the source and the receivers due to geometrical spreading, energy leakage in water and air, as well as the influence of snow. This can be achieved by exploiting the waveforms from the jumps on the ice. To this end, we proceed with the following steps:

1. isolate inversions for which source position is closest to stations $S_3$ and $S_5$, where the jumps were made;

2. calculate the corresponding energy at each geophone: $E_j^{S_i} = \int_T (U_Z(t)^2 + U_E(t)^2 + U_N(t)^2)dt$, where $T$ is the duration of the waveform, $U_Z$, $U_N$, and $U_E$ are the vertical, northward and eastward displacement components, respectively.

3. Fit the energies versus the distance, $r$, to obtain the energy decay function, $E_j(r)$. The amplitude of a guided wave in plate-like structures can be described with Hankel functions, hence we fit the square of a Hankel function to approximate this decay.

4. extrapolate the fitted energy decay function up to the largest distance between the icequakes and the geophones.

Jumps on the ice were performed from a height of one meter by a person weighing 85 kg, so the kinetic energy of the jumps is about 850 J. Hence $E_j(r)$ can be used to estimate $E_s(r = 0)$, the energy of the other sources, from $E_s(r(S_i))$, the energy measured from the waveforms at geophone $S_i$, located at a distance $r(S_i)$ from the source, such that:

$$E_s(r = 0) = \frac{E_s(r(S_i))}{E_j(r(S_i))} \times 850. \tag{1}$$

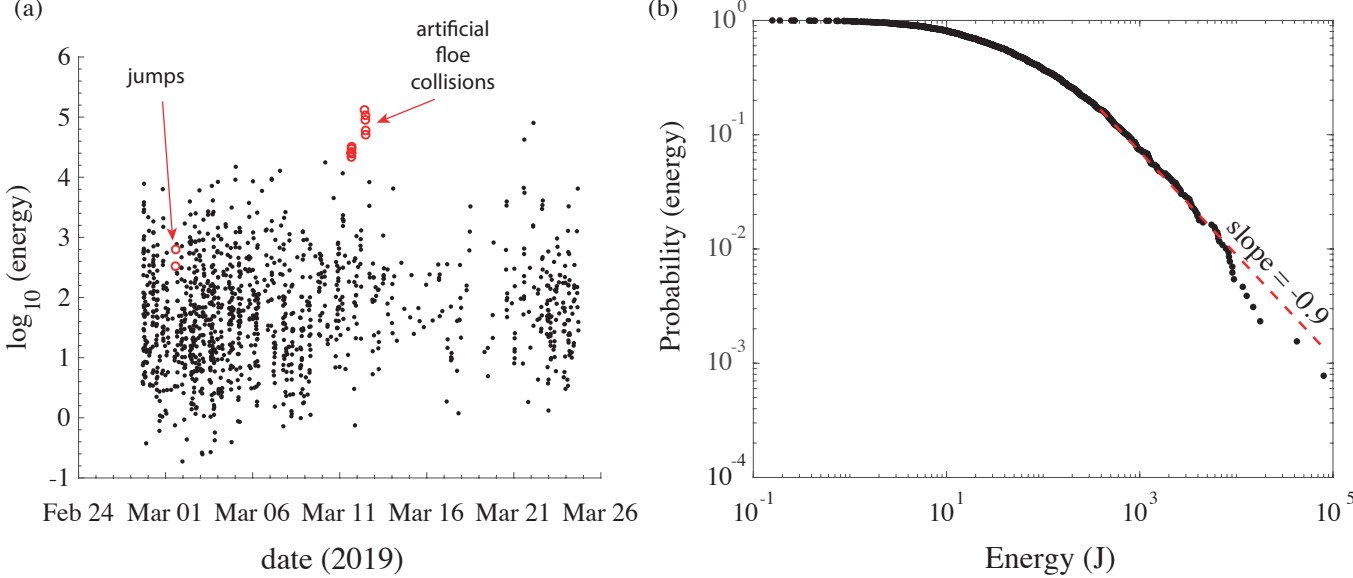

**Figure 9.** (a) energy versus date of all inverted icequakes (black dots) and artificial sources (red circles). (b) Scaling law of the icequakes energy.

For example, applying this formula to the ice floe collisions, north-east of station $S_3$ (see the black square in figure 7a), we are able to estimate the energy of these sources to be of the order of 40 kJ (figure 9a). Given the weight of the ice floe, 39 tons, the speed of the floe at the impact should be of about 1.4 m/s, which is walking speed. Of course, this approach is only a "first order" estimation of the energy, because it does not account for all the physics of the problem, such as source mechanism, source directivity, scattering, etc. However, it provides interesting statistics about the scaling law of the icequakes in sea ice.

### 3.3 Energy of the icequakes

Here we apply equation (1) to calculate the energy of the 1790 icequakes selected out of the 5350 inversions. The results are shown in terms of the energy versus date in figure 9a), and in terms of the scaling law of the energy in figure 9b. Energies vary between less than 1 J and about 40 kJ. By comparison with earthquakes energy, this corresponds to energy magnitudes between -3.7 and -0.2, where the conversion was obtained with the energy magnitude formula (Choy and Boatwright, 1995):

$$\text{magnitude} = 2/3 \log_{10}(\text{energy}) - 3.2.$$

In the artificial ice floe collisions, the impact surface is 5 m $\times$ 0.8 m $= 4$ m$^2$, and causes an energy release of about 40 kJ. The average icequakes energy is about 500 J, which is two orders of magnitude weaker. This suggests that the majority of icequakes are generated by cracks that are a few centimeters long, but there are also icequakes generated by cracks several meters long.

The scaling law of the icequakes was calculated following Clauset et al. (2009), which gave a power law with a slope of -0.9 between 400 J and 80000 J, which is validated by a Kolmogorov-Smirnov test (Massey, 1951) that gives a p-value of 0.078.

This is a bit larger than for earthquakes, for which the typical scaling law in terms of energy has a slope of -0.66 (see above in the Gutenberg-Richter law). However, the comparison is only indicative, and serves no other goal, since very little is known about sea ice dislocation mechanisms, which are presumably quite different than those of fault zones.

## 4 Discussion

The forward model used for data inversion assumes a constant ice thickness, which is not the case in reality. Our estimations of ice thickness represent an apparent value that we assume to be an average between the icequakes source and the 5 geophones. It is noteworthy that this assumption should not hold if the ice thickness varied monotonically. In that case, without a forward model that accounts for linear thickness variations, for example (Moreau et al., 2014) in a free plate, the apparent ice thickness would be biased towards the value directly under the receivers (Romeyn et al., 2021), due to adiabatic mode propagation. However, it is very unlikely that there was a monotonic thickness increase (or decrease) at the place of the experiment, although it is not possible to verify without ground-truth values. More likely would be that there were random thickness variations of a few centimeters between the shore line and the geophones. Nonetheless, the path between the source and each geophone is not the same, so the ice thickness is likely to be slightly different from one path to another. Hence the reason for the assumption that the apparent thickness is an average.

This model, like all models based on plate theory, also suffers limitation of being restricted to low frequencies. Ongoing comparisons between inversions using this model and a full numerical model based on the spectral element method (Cao et al., 2021) suggest that using a model based on plate theory underestimates the ice thickness by a few cm, as soon as the frequency band of interest includes frequencies above 10 Hz, for an ice thickness of 1 m. Moreover, these numerical investigations also reveal that the snow layer, if not accounted for in the model, lead to some estimations that reflect "apparent values" for the ice thickness and mechanical properties (Moreau et al., 2020a). In particular, the snow layer introduces a gradient of porosity through the thickness which makes the flexural wave velocity lower, resulting in potential underestimation of the ice thickness.

These are, however, preliminary results and the investigation is still ongoing. The full study will be presented in details in a separate paper. These limitations to the model (plate theory and not accounting for snow), together with the spatial heterogeneity of the ice thickness in the field, explain why the thickness estimations appear to be slightly less that those measured by drilling the ice (figure 7b).

Both the above-mentioned issues will be tackled in future developments by using the relocated icequakes as sources for a tomographic inversion of the thickness, for example based on full waveform inversion strategies with a spectral element-based forward model, which also accounts for the snow layer.

Currently, we are only making use of the vertical displacement component in this type of inversion. However, by making use of the horizontal displacement components, it will be possible to also recover and monitor Young's modulus and Poisson's ratio, instead of using a constant value. As demonstrated in Serripierri et al. (2022), these parameters can be constrained by exploiting the waveforms of the other two guided modes, $QS_0$ and $SH_0$, which polarization are on the horizontal displacement components. This could be useful on drifting pack ice for long-term monitoring. In the present study, using a constant value

for these parameters is, however, a valid assumption, since these have been shown to remain constant around $E = 3.8$ GPa and $\nu = 0.28$, during the 27 days of deployment (Serripierri et al., 2022).

A particularly appealing aspect of this approach is the ability to adapt the frequency of investigation to the required spatial resolution. The wavelength of the seismic modes guided in the ice layer depend on the product of the ice thickness by the frequency. It typically varies from a few meters around 100 Hz·m, to a few hundreds of meters around 0.1 Hz·m. Hence, on drifting pack ice, by adjusting the size of the geophones antenna, the spatial resolution of the maps can vary from a few tens of meters to a few kilometers.

The ideal monitoring conditions in the fjord allow the development of such methodologies, in view of a transfer to the open Arctic ocean. In Moreau et al. (2020b), the inversion method was successfully applied to a few icequakes identified "manually" in continuous recordings on drifting pack ice in the Arctic. To transfer the methodology of this paper in a less favorable environment, the preferred approach is to join other scientific projects on sea ice, for example during ice camps on fast ice, or onboard an icebreaker. This is planned in the coming two or three years. Another possibility, dedicated to long-term monitoring, is to make use of geophones that can telemeter the continuous recordings via satellite, such as those used in Marsan et al. (2019) to record seismic noise on drifting ice floes. However, this requires a large budget, which for the most part is dedicated to the use of satellite bandwidth.

The thermomechanical processes that generate the icequakes recorded in the fjord are likely a combination of thermal fracture such as those observed in lakes (Ruzhich et al., 2009) or in glaciers (Podolskiy et al., 2019) and mechanical forcing due to tide, as observed in glaciers (Barruol et al., 2013). The 24-hours periodicity of icequakes, as shown in figures 3 and 6, is not correlated with temperature variations, which suggests that the latter effect is dominant compared to the former. The range of energies of the recorded icequakes is consistent with those reported in frozen lakes, for example Ruzhich et al. (2009) or Kavanaugh et al. (2018).

## 5   Conclusions

We conducted a systematic analysis of the microseismicity recorded on fast ice in Svalbard. The analysis consists of a two-step processing of the seismic data. First, a deep learning algorithm is used for clustering the waveforms into different families of signals: icequakes, background noise, anthropogenic noise, etc. Clusters containing thousands of icequakes were identified, from which the waveforms of the $QS$ mode were extracted. Then, Bayesian inversion was applied to these waveforms to determine the position of the seismic sources and the average ice thickness between the sources and the geophones. Icequakes were found to originate from all along the shore line of the fjord, where mechanical stress due to tides induces cracking, most of which occur with a recurrence of about 24 hours. Our analyses also reveal that the recorded icequakes are likely to be generated by source mechanisms that are quite different from dislocations encountered for earthquakes, because the energy of the icequakes is mainly on the horizontal displacement components, which are dominated by the $SH_0$ and $QS_0$ modes. However, these modes are excited by traction/compression motion, which is not compatible with the anti-symmetric motion of vertical dislocations.

The ice thickness was found to increase by about 15 cm during the first two weeks of deployment, which is consistent with the low temperatures in the first half of March 2019, and confirmed by ice drillings at the beginning and the end of March. Finally, using artificial impulsive sources, we were able to determine a scaling law of the icequakes energy, ranging between 1J and 30 kJ and exhibiting a log-normal distribution with slope -0.9.

In a future work, by including the waveforms of the three guided modes instead of using only that of the $QS$ mode, it will be possible to exploit the whole content of the recordings via full waveform inversions strategies, in order to generate maps of sea ice parameters with a spatial resolution of a few tens of meters.

These data will also be very useful to train a deep neural network able to estimate instantly both the source position and the ice properties, without the need to go through a computationally expensive MCMC inversion, enabling the possibility of real-time in situ estimation of the ice thickness and cracks.

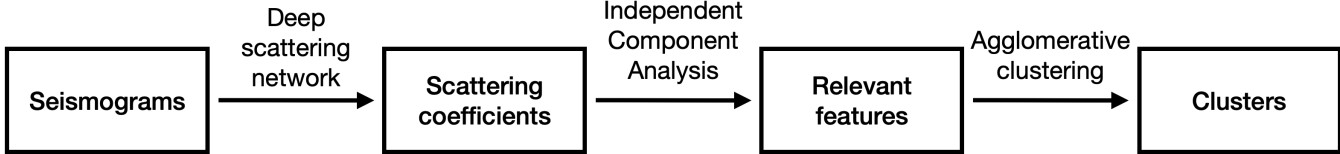

**Figure A1.** Computing scheme of the Scatseisnet method. The seismograms are fed to a deep scattering network for extracting the scattering coefficients. The most relevant features are extracted from the scattering coefficients with an independent components analysis. Clusters are identified with an agglomerative clustering technique.

## Appendix A: Scatseisnet strategy

The Scatseisnet algorithm operates with the processing steps shown in figure A1. These steps are briefly detailed in this section.

### Deep Scattering network

A scattering network is a deep convolutional neural network with wavelet filters. Although a thorough description of the deep scattering network is provided in Andén and Mallat (2014), we provide a succinct description here for self-consistency of this
paper. Considering the continuous three-component input data segment $u(t) \in \mathbb{R}^3$, the scattering coefficients $S^{(\ell)}$ of order $\ell$ are the result of a cascade of wavelet convolutions and modulus operations, such as

$$S^{(\ell)}\left(t, f_1, f_2, \ldots, f_\ell\right) = \max_{[t,t+dt]} \left|\phi^{(\ell)}\left(f_\ell\right) \star \left|\ldots \star \left|\phi^{(2)}\left(f_2\right) \star \left|\phi^{(1)}\left(f_1\right) \star x\right|\right|\right|\right|, \tag{2}$$

where $\star$ denotes the convolution operation, $|\cdot|$ is the modulus, and $\phi^{(i)}(f_i)$ is the wavelet filter at the layer $i$ of the scattering network, with center frequency $f_i$. Here $f_i$ refers to one of the center frequencies of the layer $i$, which is defined by the operator.
While the authors in Seydoux et al. (2020) implement learnable wavelet filters $\phi^{(i)}$ with respect to the clustering loss, we here directly use Gabor filters, as originally proposed in Andén and Mallat (2014), and implemented in Steinmann et al. (2022) to allow for faster computations. The maximum operator perform a pooling reduction over the time interval $[t, t+dt]$ allowing to reduce the complexity carried by the waveform itself. We prefer it over the lowpass-filter operation originally proposed in Andén and Mallat (2014) to maximize our change to make pulse-like signals dominate the final representation.

**Scattering coefficients**

The first wavelet transform provides a time frequency representation of the input seismic waveform –namely a scalogram– which is routinely used by seismologists. Thanks to the modulus operation, the wavelet transform $|\phi^{(1)} \star x|$ represent the envelop of the input signal as a function of time in the frequency band of the wavelet filter $\phi^{(1)}(f_1)$ centered around the frequency $f_1$. The second-order wavelet transforms perform a scalogram of every envelopes provided by the first-order wavelet transform.
This second-order transform provides information on the modulation of the signal's envelope within different frequency bands, and therefore allow to discriminate signals with the same frequency content but different temporal histories. Following Andén

and Mallat (2014), we use a maximum of two layers in the scattering network, provided that the third layer slightly improves auto classification performances at a high computational cost.

**Independent component analysis**

The number of coefficients generated by the deep scattering network is large (few hundreds), meaning that the dimension of the feature vector for every window is high-dimensional. Although the dimension is large, the information provided by neighboring scattering coefficient is highly similar in essence (Andén and Mallat, 2014). In order to reduce the dimension, and inherently, improve the computational performances, we extract the most relevant features with an independent component analysis (Comon, 1992), which solves the problem of factorizing the data matrix into a source matrix, and a mixing matrix

under the constrain that the sources should be maximally independent.

**Agglomerative clustering**

We finally use agglomerative clustering to identify clusters in the data. By computing the linkage matrix, we form a dendrogram (as represented in Figure 3a. The dendrogram indicates how cluster form as a function of the merging distance. By defining an arbitrary threshold, we can obtain a varying number of clusters. For more details, please consider reading Steinmann et al.

(2022).

*Author contributions.* Ludovic Moreau designed and led the Icewaveguide experiment in the scope of the IWG project (ANR10 LABX56). He supervised all this manuscript and research. Ludovic Moreau and Jérome Weiss participated in the deployment of the seismic array. Leonard Seydoux and Michel Campillo designed the Scatseisnet algorithm

*Competing interests.* The authors declare that they have no conflict of interest.

*Acknowledgements.* ISTerre is part of Labex OSUG@2020 (ANR10 LABX56). This research was funded by the Agence Nationale de la Recherche (ANR, France) and by the Institut Polaire Français Paul-Emile Victor (IPEV).

All data used for this research are from the network with FDNS code XG (Moreau and RESIF, 2019).

Most of the computations presented in this paper were performed using the GRICAD infrastructure (https://gricad.univ-grenoble-alpes.fr), which is supported by Grenoble research communities.

390  M. Campillo and L. Seydoux acknowledge support from the European Research Council under the European Union Horizon 2020 research and innovation program (grant agreement no. 742335, F-IMAGE).

M. Campillo and L. Seydoux had support from the Multidisciplinary Institute in Artificial Intelligence MIAI@Grenoble Alpes (Program "Investissements d'avenir" contract ANR-19-P3IA-0003, France).

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
