# Peer review of "Analysis of micro-seismicity in sea ice with deep learning and Bayesian inference: application to high-resolution thickness monitoring"

_The Cryosphere, 2022_

## Referee Comment (RC1)

**General Comments**

This is a nice manuscript that I enjoyed reading. The large catalogue of icequake waveforms that the authors have analysed makes a new and important contribution to the field. The authors also demonstrate a methodology to efficiently isolate these icequake signals in long-term ambient recordings that appears to work well. The authors also make a nice attempt to calibrate their measurements so that the magnitudes and rupture lengths associated with the recorded icequakes can be quantified (roughly). I think there are several aspects that can be significantly strengthened in the manuscript, mostly relating to how the results of the study are presented and interpreted. I have outlined these aspects in detail in the following sections and expect that it should be quite possible for the authors to address these with relatively minor modifications to the manuscript.

**Apparent variation in ice thickness**

It is notable that the standard deviation corresponding to thickness estimates from individual estimates is quite small, 2 cm, but the range of thicknesses estimated from multiple events during any given period is much larger at around 20 cm (shown in Figure 6b). The authors only comment on the long-term increasing trend as reflecting ice growth over the month-long experiment but do not give much attention to the spread in estimates. Do the authors think that this spread reflects actual spatial variation in ice thickness, and can this be confirmed by the ice drilling? If not, could there be some other effect that explains why the thickness estimates vary so much?

It looks like there is a trend that the ice close to the shoreline was thicker than the ice away from the shoreline (e.g. Figure 6a). Can this be confirmed as real spatial thickness variation by drilling? The apparent increase of flexural wave estimated ice thickness close to the shore is also consistent with observations of Romeyn et al. (2021). Could this be explained in terms of a finite-plate boundary condition effect as hypothesised in Appendix 1 of Romeyn et al. (2021)? According to that hypothesis, with a Poisson's ratio of 0.28, a correction factor of 0.62 should be applied to equate the thickness of a clamped plate (representing ice near the shoreline) with a simply supported plate (representing ice farther from shore) giving equal maximum tangential stresses. The thickness estimates in Figure 6 are about 0.7 m near the shoreline and $0.7 \times 0.62 = 0.43\ m$ which is strikingly consistent with the thickness estimates located further away from the shoreline. Could this be an explanation for the large spread in estimated thickness (~20cm) that is observed at a given time, as shown in Figure 6b?

To reiterate, there are several mentions of drilled thicknesses but the actual results i.e., thicknesses, and locations of these measurements are not given. These should certainly be added given the usefulness of physical thickness measurements for validating, calibrating and understanding the flexural wave thickness estimates.

**Spatial interpretation of ice thickness estimates**

The authors state on Line 246: "Our estimations of ice thickness represent an apparent value that is averaged between the icequakes source and the 5 geophones". Is this property known or is it an assumption (that the thickness estimate represents the average ice thickness between source and receiver)? To test this one would need to do a reciprocity test, i.e., does switching the source and geophone position give an identical signal over an area where the ice thickness is varying? How can we discard the possibility that the recorded signal is dominated by the ice thickness in the vicinity of the recording geophone, for example? Indeed, this would be consistent with the adiabatic wave

concept whereby the phase velocity of guided waves varies smoothly according to the local thickness as they propagate through a waveguide with a gradually varying thickness. Here are a few references that give some background on this topic:

Ech-Cherif El-Kettani, Mounsif & Luppé, Francine & Guillet, A. (2004). Guided waves in a plate with linearly varying thickness: Experimental and numerical results. Ultrasonics. 42. 807-12. 10.1016/j.ultras.2004.01.071.

El Kettani, M. C., & Hamitouche, Z. (2009). Inverse problem for the geometry profile determination of waveguides with varying section using adiabatic behavior of guided waves. IEEE transactions on ultrasonics, ferroelectrics, and frequency control, 56(9), 2023-2026.

Hu, Z., An, Z., Kong, Y., Lian, G., & Wang, X. (2019). The nonlinear S0 Lamb mode in a plate with a linearly-varying thickness. Ultrasonics, 94, 102-108.

The jumps the authors studied near stations S3 and S5 could be used to test source-receiver reciprocity, although the result will still be ambiguous if the ice thickness is constant between stations S3 and S5. The tomographic inversion technique proposed by the authors for a future study might also help to resolve this issue, but I would be careful about assuming that a simple path average is the solution based on the data that has been presented to date. Please consider this point carefully and at least re-phrase along the lines of "we assume that the estimations of ice thickness represent an apparent value that is averaged between the icequakes source and the 5 geophones".

**Interpretation of icequakes as dominantly thermally driven due to 24-hour periodicity**

I tend to disagree with the authors interpretation that the 24-hour periodicity of the recorded icequake seismicity counts against tidal stress and in favour of thermal stress as the dominant icequake source mechanism. I have given more details in the specific comment on Line 123-125, but in general the tidal forcing does have a 12/24 hr periodicity and the fact that the tidal magnitude is on the order of tens of centimetres does not necessarily mean the stresses will be insufficient to initiate cracking and produce icequakes. On the other hand, it seems straightforward to demonstrate that the air temperature during the study period does not have a 24-hour periodicity (see specific comment on Line 123-125), due to the low height of the sun and dominance of synoptic weather patterns in driving temperature variation in this region (e.g. Bednorz, 2011). It would certainly make sense to include an illustration of the air temperature timeseries in the manuscript, given that it is also mentioned in several other places relating to the timing of ice growth.

Bednorz, E. (2011). Occurrence of winter air temperature extremes in Central Spitsbergen. Theoretical and Applied Climatology, 106(3), 547-556.

The clustering of icequake seismicity around the perimeter of Vallunden Lake near the shoreline is also consistent with movement on tidal cracks, which are a typical feature of fast ice, driven by the tidal cycle. The authors may find the following reference useful, which gives further detail on stress cycling in the vicinity of tidal cracks based on measurements from Van Mijenfjorden (within a couple of kilometres from Vallunden Lake, the study area of this manuscript):

Caline, F. & Barrault, S. (2008) *Measurements of stresses in the coastal ice on both sides of a tidal crack*. In 19th IAHR International Symposium on Ice, Vancouver, British Columbia, Canada, 2008. URL: http://malemuk.com/olofee/wp-content/uploads/2015/12/Paper-070-Caline-and-Barrault.pdf

See also the definition of "tide crack" given, for example, in the McGraw-Hill Dictionary of Scientific & Technical Terms:

> "A crack in sea ice, parallel to the shore, caused by the vertical movement of the water due to tides; several such cracks often appear as a family."

> tide crack. (n.d.) McGraw-Hill Dictionary of Scientific & Technical Terms, 6E. (2003). Retrieved December 1 2022 from https://encyclopedia2.thefreedictionary.com/tide+crack

**Water depth effect on QS mode dispersion?**

It is worth noting that the authors assume a model of dispersion for the QS mode that does not account for the effect of finite water depth. The maximum water depth in Vallunden Lake appears to be ~10 m based on Marchenko et al. (2021). Using the dispersion relation of Romeyn et al. (2021) that does include the finite water depth, one can estimate that ignoring the water depth (assuming it is infinite) leads to an overestimation of phase velocity by 226% at 1 Hz, 23 % at 4 Hz and 4 % at 8 Hz. The impact on the results of this study may have been assumed to be minor since the dominant frequency of the QS energy is around 8 Hz? However, since the authors state the icequakes are associated with signals spanning the frequency range 1-50 Hz it would be reasonable to give a justification for ignoring the finite water depth in the manuscript.

> Marchenko, A.V., Morozov, E.G., Ivanov, A.V., Elizarova, T.G, Frey, D.I., (2021) *Freezing of Tidal Flow in Lake Vallunden (Spitsbergen),* Port and Ocean Engineering Under Arctic Conditions, Proceedings 2021 URL: https://www.poac.com/Proceedings/2021/POAC21-054.pdf

**Discussion**

In general, I think the discussion focusses a bit much on future research prospects without fully discussing the results of the present study and their implications. There are some interesting results presented in this study from a large catalogue of icequakes and I think that these should be focussed on a bit more since this, to me, is the most novel aspect of this study.

**Specific Comments**

Line 17-18: "…sea ice is an essential element of polar regions because of the role it plays in phytoplankton production, and in several atmosphere-ice-oceans interactions". Please add one or more references to support, particularly since the interactions are not explained here.

Line 19: "…important negative trend of about 12.6% per decade, according to the National Snow and Ice Data Center". A specific reference should be included to support this result.

Line 27-28: "…thick ice filters light more than thin ice, hence thickness influences phytoplankton production". This should be supported with a reference.

Line 28-29: "…Thicker ice is also more resilient to external forcing such as swell or wind forcing.". This should be supported with a reference.

Line 50-51: "With hundreds of icequakes recorded everyday, a daily temporal resolution can be achieved." It could be worth adding that deployment location is an important consideration for this type of monitoring. In this case the periodic tidal forcing is an important driver of icequake seismicity,

but different mechanisms may operate at other locations so that understanding the local icequake seismicity will be important for others aiming to implement this methodology.

Line 64: "Guided modes are dispersive, hence seismic signals recorded in sea ice away from the source are distorted." Only some guided wave modes are dispersive, so this sentence should be revised. The following is quoted from Moreau et al. (2020a) which this manuscript follows: *"The SH0 mode is not dispersive, and the QS0 mode becomes dispersive only at much higher frequency-thickness values (above 1000 Hz·m)."*

Line 64-66: "An important property of guided wave propagation is the one-to-one relationship between the dispersion of the waveforms, the mechanical properties of the ice, its thickness and the source-receiver distance". I suggest removing "one-to-one" from this sentence, since these properties are not necessarily independent of one another as a one-to-one relationship would imply. Alternatively, the authors should demonstrate that these properties are independent so that each possible combination gives a unique waveform. Alternatively remove "mechanical properties" since these are assumed to be constant and it is probably justifiable that there is a one-to-one relationship between dispersion of the waveforms, the ice thickness and source-receiver distance. Yet another alternative would be to mention the utility of multimodal for constraining all of these properties (as mentioned several other places in the manuscript).

Line 74: "…with cracks located for the most part along the shoreline". Don't these cracks fit quite well with the definition of tide cracks? This is given, for example, by the McGraw-Hill Dictionary of Scientific & Technical Terms:

"A crack in sea ice, parallel to the shore, caused by the vertical movement of the water due to tides; several such cracks often appear as a family."

> *tide crack. (n.d.) McGraw-Hill Dictionary of Scientific & Technical Terms, 6E. (2003). Retrieved December 1 2022 from https://encyclopedia2.thefreedictionary.com/tide+crack*

Line 89-90: "we introduced an approach based on a Bayesian inversion of the icequakes waveform to recover the ice thickness while simultaneously relocating the source position". Consider adding something like "for elastic parameters E and ν assumed *a priori*". It would be useful for the reader to keep track of which parameters are inverted for and which are assumed or held constant.

Line 121: "the associated signals have an average frequency content between 1 and 50 Hz" Average frequency content is a bit ambiguous in this context since it can be confused with the second part of the sentence dealing with the dominant or central frequency. What about "the associated signals are composed of frequencies spanning from 1 to 50 Hz"?

Line 122: "Icequakes are likely produced by thermomechanical forcing." Please add a reference here. I would suggest the following as highly relevant: Olinger, S., Lipovsky, B., Wiens, D., Aster, R., Bromirski, P., Chen, Z., Gerstoft, P., Nyblade, A. A., and Stephen, R.: Tidal and thermal stresses drive seismicity along a major Ross Ice Shelf rift, Geophys. Res. Lett., 46, 6644–6652, 2019.

Line 123-125: "the majority of icequakes occurs with a period of 24 hours (figure 5). This periodicity can also be seen in figure 3b, especially between March 1st and March 15th." It must be figure 3c that is referred to here? Moreover, the authors comment on the magnitude of the semidiurnal tide being 10-20 cm, but not on the magnitude of diurnal temperature variations, which should be simple enough to include and could be a very useful addition to the discussion of the 24-hour periodicity of icequakes and its thermomechanical interpretation. At this high northern latitude, the sun only reaches a maximum of ~10 degrees above the horizon in mid-March and it is not a given that diurnal

insolation patterns will be the main driver of temperature variations compared to the passage of synoptic weather systems.

As an example, below is the air temperature record for the nearby Sveagruva weather station (SN99760), obtained from the public database https://seklima.met.no/observations/ for March 2019. The lack of an apparent 24-hour periodicity in the temperature record indicates that, contrary to the interpretation of the authors, the tidal forcing is a more probable driver of icequake seismicity than temperature.

[Figure]

The following paper by Marchenko & Morozov (2013) would also be highly relevant to cite, since it deals exactly with the tidal cycle at the study location presented in this manuscript.

Marchenko, A. V. and Morozov, E. G.: Asymmetric tide in Lake Vallunden (Spitsbergen), Nonlin. Processes Geophys., 20, 935–944, https://doi.org/10.5194/npg-20-935-2013, 2013.

Line 178-180: "sources are located essentially along the shore line, where most of the stress is concentrated due to thermal expansion and the mechanical tension caused by tide." I think this point could be made more rigorously. The observation seems to be in very good agreement with the dynamics of tidal cracks, a commonly observed feature associated with fast ice. The authors may like to investigate other references in addition, but Caline and Barrault (2008) appears highly relevant and is also based on observations from Van Mijenfjorden.

> Caline, F. & Barrault, S. (2008) *Measurements of stresses in the coastal ice on both sides of a tidal crack*. In 19th IAHR International Symposium on Ice, Vancouver, British Columbia, Canada, 2008. URL: http://malemuk.com/olofee/wp-content/uploads/2015/12/Paper-070-Caline-and-Barrault.pdf

Line 202-203: "The ice thickness increase was also confirmed by ice drillings on March 1 and March 25." What were the drilled thicknesses and where were they measured? Please add the drilled thicknesses to Figure 6, it would be quite instructive to see how they fit with the range of estimates. Given the authors also state that the ice thickness is not constant in reality (Line 249), it is important to back this up with measurements supporting this.

Line 212: "geometrical spreading, and energy leakage in water" should be changed to geometrical spreading, and energy leakage in water and air. Probably there is also loss/dissipation of energy into the snow pack resting on the ice?

Line 273: "The 24-hours periodicity of icequakes, as shown in figures 3 and 5, suggests that the former effect is dominant compared to the latter." Similar to the earlier comment on Line 123-125 the lack of 24-hour periodicity in air temperature, and the fact that the tidal forcing does have a 12/24hr periodicity (Marchenko & Morozov, 2013) rather suggests the opposite, i.e., that tidal forcing is the dominant driver of icequake seismicity recorded in this dataset.

**Specific comments on figures**

Figure 2: Since the raw data was converted to displacements and the instrument response has been deconvolved, the units of displacement should be included in the figure.

Figure 3: Black vertical line indicating threshold distance should be annotated in the figure caption.

Figure 6: Since Figure 6a includes both spatial and temporal thickness variation it is hard to interpret. Consider adding an additional panel showing the results from one day of recording (discussed as a possibility from Line 207-209)?

**Technical Corrections**

Line 146: Acronym MCMC should be stated in full as Markov Chain Monte Carlo on first use.

---

## Author Comment (AC1)

We warmly thank the reviewer for such an in-depth review that is a significant contribution to improving the manuscript. We have answered all points in details. Please see below.

**General Comments**

This is a nice manuscript that I enjoyed reading. The large catalogue of icequake waveforms that the authors have analyzed makes a new and important contribution to the field. The authors also demonstrate a methodology to efficiently isolate these icequake signals in long-term ambient recordings that appears to work well. The authors also make a nice attempt to calibrate their measurements so that the magnitudes and rupture lengths associated with the recorded icequakes can be quantified (roughly). I think there are several aspects that can be significantly strengthened in the manuscript, mostly relating to how the results of the study are presented and interpreted. I have outlined these aspects in detail in the following sections and expect that it should be quite possible for the authors to address these with relatively minor modifications to the manuscript.

**Apparent variation in ice thickness**

It is notable that the standard deviation corresponding to thickness estimates from individual estimates is quite small, 2 cm, but the range of thicknesses estimated from multiple events during any given period is much larger at around 20 cm (shown in Figure 6b). The authors only comment on the long-term increasing trend as reflecting ice growth over the month-long experiment but do not give much attention to the spread in estimates. Do the authors think that this spread reflects actual spatial variation in ice thickness, and can this be confirmed by the ice drilling? If not, could there be some other effect that explains why the thickness estimates vary so much?

The standard deviation of each thickness estimate is related to how well the model fits the data in the MCMC inversion, hence it should, in theory, not be related to the range of values found when one looks at all positions. This should be mitigated by the fact that the inferred values integrate 5 different propagation paths, and thus to a small extent they are sensitive to spatial variations. But not to the point that it translates in the Markov Chain. To make this point more convincing, we are adding the following figure to the manuscript.

[Figure]

Figure 8 – a) Same as figure 7a, restricted to icequakes originating only from directions marked as [1], [2] and [3]. b) same as figure 7b, for the icequakes in the three groups shown in figure 8a: + are for group [1], o are for group [2] and □ are for group [3]. Inversions originating from a same region and at a similar time have a range of thicknesses that remain within the standard deviation. When comparing all directions, however, the range of thicknesses is of the order of 20 cm.

It looks like there is a trend that the ice close to the shoreline was thicker than the ice away from the shoreline (e.g. Figure 6a). Can this be confirmed as real spatial thickness variation by drilling? The

apparent increase of flexural wave estimated ice thickness close to the shore is also consistent with observations of Romeyn et al. (2021). Could this be explained in terms of a finite-plate boundary condition effect as hypothesised in Appendix 1 of Romeyn et al. (2021)? According to that hypothesis, with a Poisson's ratio of 0.28, a correction factor of 0.62 should be applied to equate the thickness of a clamped plate (representing ice near the shoreline) with a simply supported plate (representing ice farther from shore) giving equal maximum tangential stresses. The thickness estimates in Figure 6 are about 0.7 m near the shoreline and $0.7 \times 0.62 = 0.43\ m$ which is strikingly consistent with the thickness estimates located further away from the shoreline. Could this be an explanation for the large spread in estimated thickness (~20cm) that is observed at a given time, as shown in Figure 6b?

To reiterate, there are several mentions of drilled thicknesses but the actual results i.e., thicknesses, and locations of these measurements are not given. These should certainly be added given the usefulness of physical thickness measurements for validating, calibrating and understanding the flexural wave thickness estimates.

This is an interesting hypothesis. We agree that the figure may leave this impression. However, this is an artefact that comes from the fact that sources positions were plotted with respect to the date of icequakes occurrence. Since ice thickness increased with time during March 2019, sources associated with large thicknesses appear on top of hundreds of other sources associated with smaller thicknesses. To the north of the stations, for example, many sources with small thickness appear to be closer to the shore than sources with larger thicknesses. Unfortunately, we do not have drillings to verify that.

We believe that the correction factor does not apply here, since this would lead to thicknesses that are not consistent with the few drillings that we made in the field near station S1. These were reported in Moreau et al. (2020a), with exact locations, but we have indicated them in figure 1b of this manuscript and reported the corresponding thicknesses in figure 7b. Three drillings were made within a radius of 30 m around station S1. One was performed on March 1, giving a thickness of 62 cm, and two were performed on March 26, giving thicknesses of 70 and 73 cm. It is worth mentioning that Marchenko et al. (2021), also reported an ice thickness of about 80 cm on March 11 near the area marked with a black square in figure 6a.

**Spatial interpretation of ice thickness estimates**

The authors state on Line 246: "Our estimations of ice thickness represent an apparent value that is averaged between the icequakes source and the 5 geophones". Is this property known or is it an assumption (that the thickness estimate represents the average ice thickness between source and receiver)? To test this one would need to do a reciprocity test, i.e., does switching the source and geophone position give an identical signal over an area where the ice thickness is varying? How can we discard the possibility that the recorded signal is dominated by the ice thickness in the vicinity of the recording geophone, for example? Indeed, this would be consistent with the adiabatic wave concept whereby the phase velocity of guided waves varies smoothly according to the local thickness as they propagate through a waveguide with a gradually varying thickness. Here are a few references that give some background on this topic:

Ech-Cherif El-Kettani, Mounsif & Luppé, Francine & Guillet, A. (2004). Guided waves in a plate with linearly varying thickness: Experimental and numerical results. Ultrasonics. 42. 807-12. 10.1016/j.ultras.2004.01.071.

El Kettani, M. C., & Hamitouche, Z. (2009). Inverse problem for the geometry profile determination of waveguides with varying section using adiabatic behavior of guided waves. IEEE transactions on ultrasonics, ferroelectrics, and frequency control, 56(9), 2023-2026.

Hu, Z., An, Z., Kong, Y., Lian, G., & Wang, X. (2019). The nonlinear S0 Lamb mode in a plate with a linearly-varying thickness. Ultrasonics, 94, 102-108.

The jumps the authors studied near stations S3 and S5 could be used to test source-receiver reciprocity, although the result will still be ambiguous if the ice thickness is constant between stations S3 and S5. The tomographic inversion technique proposed by the authors for a future study might also help to resolve this issue, but I would be careful about assuming that a simple path average is the solution based on the data that has been presented to date. Please consider this point carefully and at least re-phrase along the lines of "we assume that the estimations of ice thickness represent an apparent value that is averaged between the icequakes source and the 5 geophones".

As stated by the reviewer, adiabatic mode propagation relates to phase velocity. Using this concept to estimate the ice thickness is therefore adapted for example in the case of the air-coupled flexural wave (Romeyn et al., 2021), or for guided waves in a free plate where local wavenumbers (and thus phase velocities) can be extracted such as in El Kettani, M. C., & Hamitouche (2009).

In the approach of the manuscript, however, we make use of the group velocity. The inversion is based on the arriving time of all frequencies, not just one. Hence it is not trivial to answer the reviewer's question. In Moreau et al. (2014), (now a new reference in the manuscript) it was shown that the dispersion curves of the modes propagating in a plate with a linear thickness variation can be fitted with those obtained using a forward model that accounts for the linear variations of the thickness. The fit is obtained when the thickness in the model corresponds to that directly at the center of the array of receivers.

In the present case, a model of constant thickness is used, so a definite answer to this question cannot be given without a dedicated and thorough study, which falls out of the scope of this paper, especially considering that we have started moving towards a full numerical model in order to not be limited by the low-frequency approximation or by mechanical and topological variations of the ice. We can, however, give a partial answer to the reviewer by including the results of a few inversions made using synthetic signals obtained in floating ice with a linear thickness variation, with a spectral element-based model.

[Figure]

Figure 9 – Two profiles of ice thickness variations. Source position (vertical load) is shown as a star, and three receivers are shown as triangles.

We have considered two profiles of ice thickness. One where the thickness is decreasing, and one where the thickness is increasing, as shown in figure 9.  The waveforms obtained from these models at the receivers' positions are shown in figure 10 for the decreasing ice thickness case and in figure 11 for the increasing ice thickness case. These waveforms are compared with waveforms obtained with a floating ice sheet of constant thickness equal i) to the average thickness between the source and the

receiver on the one hand, and ii) to the thickness value that generates the best fit with the waveforms on the other hand.

The results of this numerical investigation are shown in tables 1 and 2. They indicate that waveforms from a floating ice sheet with linear thickness variation are most of the time slightly closer to waveforms obtained from a floating ice sheet with a constant thickness equal to that under the receiver than they are to waveforms obtained from floating ice sheet with a constant thickness equal the average thickness between the source and the receiver.

|  | Thickness under receiver | Average thickness between source and receiver | Constant thickness providing best fit |
|---|---|---|---|
| Receiver 1 | 86 | 89 | 85 |
| Receiver 2 | 75 | 83 | 78 |
| Receiver 3 | 64 | 78 | 70 |
| Table 1 – parameters of simulations for a floating ice sheet with a decreasing thickness | | | |

|  | Thickness under receiver | Average thickness between source and receiver | Constant Thickness providing best fit |
|---|---|---|---|
| Receiver 1 | 62 | 59 | 65 |
| Receiver 2 | 75 | 65 | 71 |
| Receiver 3 | 88 | 72 | 78 |
| Table 2 – parameters of simulations for a floating ice sheet with an increasing thickness | | | |

[Figure]

Figure 10 – Waveforms from a spectral element-based forward model of a floating ice layer on an infinite water column. Black solid lines: waveforms at the position of the three receivers located 50, 150 and 250 m away from the source in a floating ice layer with a decreasing thickness as shown in figure 9 (black solid line). Red solid lines: waveforms obtained in a plate which constant thickness is equal to the average thickness between the source and the receiver. Blue dotted lines: waveforms obtained in a plate which constant thickness returns the best fit with the waveforms from the decreasing ice thickness.

[Figure]

Figure 11 – Same as figure 10 with an increasing ice thickness.

This is, however, specific to a case where ice thickness varies monotonically. In the fjord, however, it is more likely that thickness variations are both increasing and decreasing along the wave propagation paths. In that case, the effects of increasing and decreasing thickness cancel each other. In the revised manuscript, this is now discussed, without going as deep as the analysis shown here, in order not to blur the main message of the paper. Here is what we have modified in the Discussion section.

**4 Discussion**

The forward model used for data inversion assumes a constant ice thickness, which is not the case in reality. Our estimations of ice thickness represent an apparent value that we assume to be an average between the icequakes source and the 5 geophones. It is noteworthy that this assumption should not hold if the ice thickness varies monotonically. In that case, without a
265 forward model that accounts for linear thickness variations, for example (Moreau et al., 2014) in a free plate, the apparent ice thickness would be biased towards the value directly under the receivers (Romeyn et al., 2021). It is very unlikely that there was a monotonic thickness increase (or decrease) at the place of the experiment, although it is not possible to verify without ground-truth values. More likely would be that there were random thickness variations of a few centimeters between the shore line and the geophones. Moreover, the path between the source and each geophone is not the same, so the ice thickness is likely
270 to be slightly different from one path to another. Hence the assumption that the the apparent thickness is an average.

This model, like all models based on plate theory, also suffers limitation of being restricted to low frequencies. Ongoing comparisons between inversions using this model and a full numerical model based on the spectral element model (Cao et al., 2021) suggest that using a model based on plate theory underestimates the ice thickness by a few cm, as soon as the frequency band of interest includes frequencies above 10 Hz, for an ice thickness of 1 m. These are, however, preliminary results and the
275 investigation is still ongoing. The full study will be presented in details in a separate paper.

Both the above-mentioned issues will be tackled in future developments by using the relocated icequakes as sources for a tomographic inversion of the thickness, for example based on full waveform inversion strategies with a spectral element-based forward model, which also accounts for the snow layer.

**Interpretation of icequakes as dominantly thermally driven due to 24-hour periodicity**

I tend to disagree with the authors interpretation that the 24-hour periodicity of the recorded icequake seismicity counts against tidal stress and in favour of thermal stress as the dominant icequake source mechanism. I have given more details in the specific comment on Line 123-125, but in general the tidal forcing does have a 12/24 hr periodicity and the fact that the tidal magnitude is on the order of tens of centimetres does not necessarily mean the stresses will be insufficient to initiate cracking and produce icequakes. On the other hand, it seems straightforward to demonstrate that the air temperature during the study period does not have a 24-hour periodicity (see specific comment on Line 123-125), due to the low height of the sun and dominance of synoptic weather patterns in driving temperature variation in this region (e.g. Bednorz, 2011). It would certainly make sense to include an illustration of the air temperature timeseries in the manuscript, given that it is also mentioned in several other places relating to the timing of ice growth.

Bednorz, E. (2011). Occurrence of winter air temperature extremes in Central Spitsbergen. Theoretical and Applied Climatology, 106(3), 547-556.

The clustering of icequake seismicity around the perimeter of Vallunden Lake near the shoreline is also consistent with movement on tidal cracks, which are a typical feature of fast ice, driven by the tidal cycle. The authors may find the following reference useful, which gives further detail on stress cycling in the vicinity of tidal cracks based on measurements from Van Mijenfjorden (within a couple of kilometres from Vallunden Lake, the study area of this manuscript):

Caline, F. & Barrault, S. (2008) *Measurements of stresses in the coastal ice on both sides of a tidal crack*. In 19th IAHR International Symposium on Ice, Vancouver, British Columbia, Canada, 2008. URL: http://malemuk.com/olofee/wp-content/uploads/2015/12/Paper-070- Caline-and-Barrault.pdf

See also the definition of "tide crack" given, for example, in the McGraw-Hill Dictionary of Scientific & Technical Terms:

"A crack in sea ice, parallel to the shore, caused by the vertical movement of the water due to tides; several such cracks often appear as a family."

tide crack. (n.d.) McGraw-Hill Dictionary of Scientific & Technical Terms, 6E. (2003). Retrieved December 1 2022 from https://encyclopedia2.thefreedictionary.com/tide+crack

We agree with the reviewer, but we could not find evidence that a 24h period could be associated with tidal cracks, since we were expecting a semidiurnal periodicity. The manuscript is now modified and we have added the reference to Caline and Barrault (2008), as well as that to Marchenko et al (2013) that could explain the specific 24h period at Vallunden.

135  be seen in figure 3c, especially between March 1st and March 15th. One would expect semidiurnal tide to reflect in the periodicity of the icequakes, but the specific geometry of the moraines around the experiment, together with the small channel that connects it to the fjord, generates some nonlinear effects that causes the tide in Vallunden to be asymmetric (Marchenko and Morozov, 2013). This could explain why occurrences are dominated by a period of 24h instead of 12h.

195  Figure 7a shows the map of the inversions that meet the quality threshold. One can see that sources are located essentially along the shore line, where most of the stress is concentrated. This is consistent with previous reports on the dynamics of tidal cracks. See for example the observation in the Van Mijen fjord by Caline and Barrault (2008).

**Water depth effect on QS mode dispersion?**

It is worth noting that the authors assume a model of dispersion for the QS mode that does not account for the effect of finite water depth. The maximum water depth in Vallunden Lake appears to be ~10 m based on Marchenko et al. (2021). Using the dispersion relation of Romeyn et al. (2021) that does include the finite water depth, one can estimate that ignoring the water depth (assuming it is infinite) leads to an overestimation of phase velocity by 226% at 1 Hz, 23 % at 4 Hz and 4 % at 8 Hz. The impact on the results of this study may have been assumed to be minor since the dominant frequency of the QS energy is around 8 Hz? However, since the authors state the icequakes are associated with signals spanning the frequency range 1-50 Hz it would be reasonable to give a justification for ignoring the finite water depth in the manuscript.

Marchenko, A.V., Morozov, E.G., Ivanov, A.V., Elizarova, T.G, Frey, D.I., (2021) *Freezing of Tidal Flow in Lake Vallunden (Spitsbergen),* Port and Ocean Engineering Under Arctic Conditions, Proceedings 2021 URL: https://www.poac.com/Proceedings/2021/POAC21- 054.pdf

On the vertical channel, dominated by the QS mode, the amplitude of the spectrum of icequake waveforms remains (on average) over -30 dB between 1 and 35 Hz, with a peak value around 8 Hz. On the horizontal channels, where the QS0 and SH0 modes are dominant, the spectrum remains over -30 dB up to 50 Hz (this is now indicated in the manuscript). We assume that ignoring the finite water depth of 10 m has a negligible effect on the inverted thickness, based on a comparison between the model used in the manuscript by Stein et al. (1998) and the model by Romeyn et al. (2021). See for example the following figure. Wavenumbers are almost identical in the frequency range of interest.

[Figure]

Wavenumber vs frequency for the QS mode, calculated in a 60 cm-thick ice sheet floating on water. Blue solid line: model by Stein et al. (1998) based on an infinite water depth. Black dashed line: model by Romeyn et al. (2021), based on water with a 10 m depth.

We have added the following sentence, at the beginning of section 3.1 to explain this.

165    1. given a set of parameters for source position around the array (latitude and longitude), source activation time, and ice thickness, generate the synthetic waveforms of the $QS$ mode at the geophones. Synthetic waveforms are generated based on a Ricker wavelet that is propagated in the ice using the analytical, low-frequency asymptotic model by Stein et al. (1998), with the following ice mechanical properties: Young's modulus = 3.8 GPa, Poisson's ratio = 0.28, and density = 910 kg/m$^3$ (Serripierri et al., 2022). This model cannot account for the finite water depth of about 10 m, like the

170    model by Romeyn et al. (2021) can, but by comparing both models, we have checked that this has negligible effect at the frequencies of interest.

**Discussion**

In general, I think the discussion focusses a bit much on future research prospects without fully discussing the results of the present study and their implications. There are some interesting results presented in this study from a large catalogue of icequakes and I think that these should be focused on a bit more since this, to me, is the most novel aspect of this study.

We have modified the discussion, which now focuses more on the limitations of our approach and on the average thickness hypothesis.

**Specific Comments**

Line 17-18: "...sea ice is an essential element of polar regions because of the role it plays in phytoplankton production, and in several atmosphere-ice-oceans interactions". Please add one or more references to support, particularly since the interactions are not explained here.

done

Line 19: "...important negative trend of about 12.6% per decade, according to the National Snow and Ice Data Center". A specific reference should be included to support this result.

Unfortunately, we could not find another reference to this statement. But surely the National Snow and Ice Data Center is to be trusted.

Line 27-28: "...thick ice filters light more than thin ice, hence thickness influences phytoplankton production". This should be supported with a reference.

done

Line 28-29: "...Thicker ice is also more resilient to external forcing such as swell or wind forcing.". This should be supported with a reference.

done

Line 50-51: "With hundreds of icequakes recorded everyday, a daily temporal resolution can be achieved." It could be worth adding that deployment location is an important consideration for this type of monitoring. In this case the periodic tidal forcing is an important driver of icequake seismicity, but different mechanisms may operate at other locations so that understanding the local icequake seismicity will be important for others aiming to implement this methodology.

We have added the following sentence.

the ice thickness. We demonstrate the possibility of generating maps of sea ice thickness and microseismic activity, with a
50 temporal resolution that is directly linked to the amount of icequakes recorded. In the specific configuration at the fjord, ice-quakes occurrences are driven by tide. On drifting ice, icequakes are generated by other mechanisms such as swell or ice motion, and many icequakes are also present in the ambient seismic field (Moreau et al., 2020b). With hundreds of icequakes recorded everyday, a daily temporal resolution can be achieved. We also use the energy information to calculate the scaling law of icequakes in terms of their released energy.

Line 64: "Guided modes are dispersive, hence seismic signals recorded in sea ice away from the source are distorted." Only some guided wave modes are dispersive, so this sentence should be revised. The

following is quoted from Moreau et al. (2020a) which this manuscript follows: *"The SH0 mode is not dispersive, and the QS0 mode becomes dispersive only at much higher frequency- thickness values (above 1000 Hz·m)."*

Line 64-66: "An important property of guided wave propagation is the one-to-one relationship between the dispersion of the waveforms, the mechanical properties of the ice, its thickness and the source-receiver distance". I suggest removing "one-to-one" from this sentence, since these properties are not necessarily independent of one another as a one-to-one relationship would imply. Alternatively, the authors should demonstrate that these properties are independent so that each possible combination gives a unique waveform. Alternatively remove "mechanical properties" since these are assumed to be constant and it is probably justifiable that there is a one-to-one relationship between dispersion of the waveforms, the ice thickness and source-receiver distance. Yet another alternative would be to mention the utility of multimodal for constraining all of these properties (as mentioned several other places in the manuscript).

Both the above remarks have been accounted for by changing the text as follows

> The $QS$ is highly dispersive at low frequencies, hence seismic signals recorded in sea ice away from the source are distorted. It is noteworthy that the $SH_0$ mode is not dispersive and that the $QS_0$ becomes dispersive only at higher frequencies. An important property of guided wave propagation is the one-to-one relationship between the dispersion of the waveforms, the waveguide thickness and the source-receiver distance, given a set of mechanical properties. By recording the seismic wavefield

Line 74: "...with cracks located for the most part along the shoreline". Don't these cracks fit quite well with the definition of tide cracks? This is given, for example, by the McGraw-Hill Dictionary of Scientific & Technical Terms:

"A crack in sea ice, parallel to the shore, caused by the vertical movement of the water due to tides; several such cracks often appear as a family."

*tide crack. (n.d.) McGraw-Hill Dictionary of Scientific & Technical Terms, 6E. (2003). Retrieved December 1 2022 from https://encyclopedia2.thefreedictionary.com/tide+crack*

We agree with the reviewer and the manuscript now mentions tide cracks, although the above reference was not included, since it is not a scientific paper.

Line 89-90: "we introduced an approach based on a Bayesian inversion of the icequakes waveform to recover the ice thickness while simultaneously relocating the source position". Consider adding something like "for elastic parameters E and ν assumed *a priori*". It would be useful for the reader to keep track of which parameters are inverted for and which are assumed or held constant.

We have modified the sentence:

> In Moreau et al. (2020b), we introduced an approach based on a Bayesian inversion of the icequakes waveform to recover the ice thickness while simultaneously relocating the source position, after the Young's modulus and Poisson's ratio of the ice were
> 95   estimated from noise interferometry. This method was validated on a few icequakes recorded in fast ice and in pack ice. In this

Line 121: "the associated signals have an average frequency content between 1 and 50 Hz" Average frequency content is a bit ambiguous in this context since it can be confused with the second part of the sentence dealing with the dominant or central frequency. What about "the associated signals are composed of frequencies spanning from 1 to 50 Hz"?

We have modified this part as follows

 day with the same temporal distribution, except around 9 AM where occurrences are slightly decreased (figure 3d). Figure 3e indicates that the signals have a frequency content ranging between 1 and 50 Hz. To be more specific, on the vertical channel, dominated by the $QS$ mode, the amplitude of the spectrum of icequake waveforms remains (on average) over -30 dB between 1 and 35 Hz, with a peak value around 8 Hz. On the horizontal channels, where the $QS_0$ and $SH_0$ modes are dominant, the spectrum remains over -30 dB up to 50 Hz.

Line 122: "Icequakes are likely produced by thermomechanical forcing." Please add a reference here. I would suggest the following as highly relevant: Olinger, S., Lipovsky, B., Wiens, D., Aster, R., Bromirski, P., Chen, Z., Gerstoft, P., Nyblade, A. A., and Stephen, R.: Tidal and thermal stresses drive seismicity along a major Ross Ice Shelf rift, Geophys. Res. Lett., 46, 6644–6652, 2019.

This reference was added.

Line 123-125: "the majority of icequakes occurs with a period of 24 hours (figure 5). This periodicity can also be seen in figure 3b, especially between March 1st and March 15th." It must be figure 3c that is referred to here? Moreover, the authors comment on the magnitude of the semidiurnal tide being 10-20 cm, but not on the magnitude of diurnal temperature variations, which should be simple enough to include and could be a very useful addition to the discussion of the 24-hour periodicity of icequakes and its thermomechanical interpretation. At this high northern latitude, the sun only reaches a maximum of ~10 degrees above the horizon in mid-March and it is not a given that diurnal insolation patterns will be the main driver of temperature variations compared to the passage of synoptic weather systems.

Yes this is figure 3c, thank you for pointing this out. We have now changed our interpretation of the original of icequakes to meet the reviewer's remark (please see next answer).

As an example, below is the air temperature record for the nearby Sveagruva weather station (SN99760), obtained from the public database https://seklima.met.no/observations/ for March 2019. The lack of an apparent 24-hour periodicity in the temperature record indicates that, contrary to the interpretation of the authors, the tidal forcing is a more probable driver of icequake seismicity than temperature.

Thank you for pointing to these temperatures data! We have added this figure to the manuscript.

[Figure]

The following paper by Marchenko & Morozov (2013) would also be highly relevant to cite, since it deals exactly with the tidal cycle at the study location presented in this manuscript.

Marchenko, A. V. and Morozov, E. G.: Asymmetric tide in Lake Vallunden (Spitsbergen), Nonlin. Processes Geophys., 20, 935–944, https://doi.org/10.5194/npg-20-935-2013, 2013.

We thank the reviewer for this remark, which we fully agree with. We have revised our interpretation of the 24h periodicity of the icequakes, as follows:

130     Icequakes are produced by thermomechanical forcing (Olinger et al., 2019). The temperature log can be extracted from the Sveagruva weather station (SN99760), located 2 km west of the place of experiment. These temperatures are shown in figure 5. The absence of a periodic pattern in temperature variations suggests that tides have more effect on icequakes than changes in temperature. The majority of icequakes occurs with a period of 24 hours (figure 6). This periodicity can also be seen in figure 3c, especially between March 1st and March 15th. One would expect semidiurnal tide to reflect in the periodicity of the

135     icequakes, but the specific geometry of the moraines around the experiment, together with the small channel that connects it to the fjord, generates some nonlinear effects that causes the tide in Vallunden to be asymmetric (Marchenko and Morozov, 2013). This could explain the reason why occurrences are dominated by a period of 24h instead of 12h.

Line 178-180: "sources are located essentially along the shore line, where most of the stress is concentrated due to thermal expansion and the mechanical tension caused by tide." I think this point could be made more rigorously. The observation seems to be in very good agreement with the dynamics of tidal cracks, a commonly observed feature associated with fast ice. The authors may like to investigate other references in addition, but Caline and Barrault (2008) appears highly relevant and is also based on observations from Van Mijenfjorden.

Caline, F. & Barrault, S. (2008) *Measurements of stresses in the coastal ice on both sides of a tidal crack*. In 19th IAHR International Symposium on Ice, Vancouver, British Columbia, Canada, 2008. URL: http://malemuk.com/olofee/wp-content/uploads/2015/12/Paper-070- Caline-and-Barrault.pdf

The reviewer is right and we have added this reference.

Figure 7a shows the map of the inversions that meet the quality threshold. One can see that sources are located essentially along the shore line, where most of the stress is concentrated. This is consistent with previous reports on the dynamics of tidal

195     cracks. See for example the observation in the Van Mijen fjord by Caline and Barrault (2008)

Line 202-203: "The ice thickness increase was also confirmed by ice drillings on March 1 and March 25." What were the drilled thicknesses and where were they measured? Please add the drilled thicknesses to Figure 6, it would be quite instructive to see how they fit with the range of estimates. Given the authors also state that the ice thickness is not constant in reality (Line 249), it is important to back this up with measurements supporting this.

Drillings positions and corresponding thickness values now appear in figure 1b. They values now also appear in figure 7b (previously figure 6b).

Line 212: "geometrical spreading, and energy leakage in water" should be changed to geometrical spreading, and energy leakage in water and air. Probably there is also loss/dissipation of energy into the snow pack resting on the ice?

Done

**3.2    Energy of the artificial sources**

Estimating the energy of the icequakes requires information about the decay of amplitude between the source and the receivers

230     due to geometrical spreading, energy leakage in water and air, as well as the influence of snow. This can be achieved by exploiting the waveforms from the jumps on the ice. To this end, we proceed with the following steps:

Line 273: "The 24-hours periodicity of icequakes, as shown in figures 3 and 5, suggests that the former effect is dominant compared to the latter." Similar to the earlier comment on Line 123-125 the lack of 24-hour periodicity in air temperature, and the fact that the tidal forcing does have a 12/24hr periodicity (Marchenko & Morozov, 2013) rather suggests the opposite, i.e., that tidal forcing is the dominant driver of icequake seismicity recorded in this dataset.

Correct. This has been modified in the manuscript.

**Specific comments on figures**

Figure 2: Since the raw data was converted to displacements and the instrument response has been deconvolved, the units of displacement should be included in the figure.

After double-checking this, it appears that there was a mistake in our response to the editor who asked a similar question about instrument deconvolution and data conversion. We apologize for this confusion.

We converted the raw data into miniseed format using the Fairfield software, but without Instrument response deconvolution, since it is not necessary for our methodology. Also, the data are expressed in mV, but could be converted to a velocity by dividing the waveforms data by the proportionality factor 89 V/m/s, and further converted to displacement by integration with respect to time. However, this is not necessary either for our methodology. This is now explained in the manuscript.

Figure 3: Black vertical line indicating threshold distance should be annotated in the figure caption.

done

Figure 6: Since Figure 6a includes both spatial and temporal thickness variation it is hard to interpret. Consider adding an additional panel showing the results from one day of recording (discussed as a possibility from Line 207-209)?

We have added a new figure (now figure 8) which is introduced in the response to the question about the standard deviation of the thickness estimations.

**Technical Corrections**

Line 146: Acronym MCMC should be stated in full as Markov Chain Monte Carlo on first use.

Done

---

## Author Comment (AC2)

**General Comments**

The paper uses machine learning to identify and extract flexural-gravity (FG) wave signals recorded on a small lake, and then uses these signals for an MCMC inversion of ice thickness, based on the dispersive nature of FG waves.

I have only passing knowledge of ML and thus cannot evaluate those portions of the paper. Details on the MCMC inversion were specified in another paper, which I did not track down, and so I also cannot evaluate the specifics of that. The core results of the ML clustering and MCMC inversion, as presented, do appear realistic. FG wave sources are back-located to the grounding lines of the lake ice and are likely icequakes. Lake ice thickens during the first two weeks of the study period, and stabilizes during the last two, consistent with stated (but not provided) temperature readings.

The manuscript has a marked lack of supporting ancillary data, specifically temperature and tidal observations. Despite this, the authors make a geophysical interpretation of their icequake as being thermally-driven, as opposed to tidally-driven. As outlined in the specific comments below, based on publicly-available temperature and tidal data, I disagree with this interpretation.

The manuscript mentions that it produces results that are in agreement with another study by the same group of authors (Serripierri et al., 2022). From a quick readthrough of that work, it appears that it uses the same methods, but with greater rigor and scope. It mentions a future publication (assumed to be the current manuscript) that will attempt to replicate their results using fewer stations and wave modes. In that context, the current work appears to be a companion paper, yet does not address the work done by Serripierri et al. (2022), or make any explicit statement of the differences between the two works. The current manuscript should make a clear statement on how it is substantially different from the prior work.

I believe the core results (ML extraction of events and MCMC inversion of FG waves) are sound, novel, and notable. However, I recommend revisions to address the points raised in the specific comments below. In particular, the lack of ancillary data and the related geophysical interpretation, and the implications for an assumption of an infinite-depth water column in a borderline shallow-water setting.

We thank the reviewer for such an in-depth review, and in particular for the help in interpreting the icequakes occurrences in conjunction with tides. We have accounted for all the comments (see detailed answers thereafter), and modified the manuscript accordingly.

A quick note about the comparison between the method in Serripierri et al. (2022) and that in the manuscript. In Serripierri et al. (2022), a frequency vs wavenumber analysis is performed via a Fourier transform on the time and space dimensions. The inversion is based on wavenumber inversion (phase velocity). This requires a dense line of geophones for spatial sampling. In the present paper, we use waveforms inversion, which is a very different approach, which allows to recover the ice thickness with only 5 stations instead of 50, which is an order of magnitude less. However, thickness values are consistent.

**Specific Comments**

Lines 11, 70, 92 : The authors state that they installed their seismic array on sea ice in Van Mijen fjord. Figure 1 shows that the field site is actually Vallunden Lake. While there is a short (100 m) channel connecting the lake to the fjord, the lake is geophysically distinct from the fjord. The authors do not provide any geologic context for the lake nor provide relevant references. The depth of Lake Vallunden

is ~10 m (Marchenko et al. 2013), which is also not mentioned by the authors, but is an important consideration for their modeling (addressed further below).

We have added the reference Marchenko and Morozov (2013) to provide the depth of the Lake, and we have more specifically explained that the location of the experiment is at Vallunden.

> To record the seismic wavefield in sea ice, an experiment was conducted on fast ice in Svalbard (Norway), in a specific part
> 75 of the Van Mijen Fjord called Vallunden (Figure 1a). This part of the fjord is surrounded by moraines, and can therefore be regarded as a "lake connected to the fjord", with a depth of about 10 m (Marchenko and Morozov, 2013). A dense array of 247

Line 13 : "calibrated seismic sources". The seismic sources, as described on line 221, appear to be more "estimated" than "calibrated." Addressed further below.

The height of the jumps was 1m, but unfortunately we cannot be as accurate as to go down to the cm. We have replaced "calibrated" occurrences by "artificial".

Lines 18–19 : Citation needed. https://nsidc.org/arcticseaicenews/charctic-interactive-sea-ice-graph/ ?

This cannot be cited as a reference paper, so we have added the hyperlink to the text.

Lines 31–33 : The densities of snow and ice are also a source of error for seismic measurements, so this statement is a little misleading in the context of putting forth seismology as an alternative to freeboard measurements. In my opinion, the greatest advantage of seismic methods vs satellite is orders of magnitude greater spatial and temporal resolutions, at the expense of spatial scale. The authors appear to agree with this, though not explicitly.

In Serripierri et al (2022), we demonstrated that the density of sea ice can be accurately evaluated through passive seismic methods, and this is one important novelty of the approaches that we develop. We have modified this part of the introduction to remind that density can be monitored as well.

> Therefore, over the past decade, there has been a renewed interest in seismic methods as a complementary means of monitoring the thickness, density and elastic properties of sea ice (Marsan et al., 2012; Moreau et al., 2020a, b; Romeyn et al., 2021; Serripierri et al., 2022).

It is true that we have not yet demonstrated that the density of snow can be constrained too. The impact of this parameter on our inversions is very minor at the frequencies of interest. However, we intend to make use of the higher frequency content to constrain the snow properties, by using a forward model that accounts for the snow cover. This is out of the scope of this paper, but a new paper where we investigate this issue will be submitted in the coming months.

Line 35 : "The first seismic experiments on sea ice date back to the late 1950s…" Ewing and Crary (1934) on Saylor's Lake and Crary (1954) on Fletcher's Ice Island should not be overlooked.

We thank the reviewer for these references, which we were unaware of. We have added reference Crary (1954) to the manuscript. However, we have not added the other reference, because it does not concern sea ice.

Lines 122, 273–274, 283–284, Fig 5 caption : "Icequakes (in the 0–7 cluster family] are likely produced by thermomechanical forcing." This interpretation is not substantiated by the data presented.

The reviewer is right, and since thermomechanical cannot be completely ruled out, we have modified this sentence such that: "Icequakes are produced by \hl{tidal and} thermomechanical forcing".

Hourly temperature data are not included in this manuscript, nor any of the references that I checked. Hourly temperatures recorded at Lufthavn, 50 km away, for March 2019 (https://meteostat.net/en/station/01008) do not show a diurnal cycle, consistent with a perpetually overcast Arctic coastal climate where temperatures are dominated by weather rather than solar heating. In the absence of locally recorded data, one could assume that Lake Vallunden has similar temperature trends. The noted 24-hour peak in icequake occurrence cannot be attributed to thermomechanical fracturing without an hourly-scale time domain correlation between temperature at Vallunden and icequake occurrence rates. Arguably, a spatiotemporal correlation would be most appropriate.

Fig. 3d does not show a consistent 24-hour recurrence pattern for the 0–7 cluster family. The deficit at 9 AM (local? UTC?) can be attributed to tides, as explained below.

We agree with the reviewer, and we have added a new figure with the temperatures recorded in March 2019 at Sveagruva (~ 1.5 km from the place of the experiement). Reviewer 1 also pointed out this issue. We have modified the manuscript to explain that icequakes are more likely a consequence of tidal forcing than temperature changes.

> be seen in figure 3c, especially between March 1st and March 15th. One would expect semidiurnal tide to reflect in the peri-
> 135 odicity of the icequakes, but the specific geometry of the moraines around the experiment, together with the small channel that connects it to the fjord, generates some nonlinear effects that causes the tide in Vallunden to be asymmetric (Marchenko and Morozov, 2013). This could explain why occurrences are dominated by a period of 24h instead of 12h.

> 195     Figure 7a shows the map of the inversions that meet the quality threshold. One can see that sources are located essentially along the shore line, where most of the stress is concentrated. This is consistent with previous reports on the dynamics of tidal cracks. See for example the observation in the Van Mijen fjord by Caline and Barrault (2008).

Fig. 5 : In the absence of any collaborating geophysical data to the contrary, I would suspect that the n*24-hour peaks are binning-related artifacts.

Now that the manuscript has been modified to correlate the icequakes with tides, it appears that these peaks are actually associated to tides that repeat every ~ n*24-hour

Fig. 6 indicates that the icequakes in the 0–7 cluster family were overwhelmingly back-located to the lake ice grounding zones. This is more suggestive of tidally forced fracture (e.g., Cole, 2020). One would expect thermomechanical fracturing to occur uniformly distributed throughout the interior of the lake ice. An argument could be made for solar heating of exposed geology at the shorelines, but would require in situ temperature and solar radiance data to validate.

We agree with the reviewer, and this was also pointed out by reviewer 1. The manuscript was modified accordingly and a reference to tidal cracks was added.

Lines 122–123, 273–274 : "[S]emidiurnal tide reaches 10-20 cm, so it is likely that tides have less effect than changes in temperature…." The opposite interpretation is suggested by the data presented. The stated tidal range is 0.1–0.2 m, potentially 30–40% of the 0.45–0.6 m inverted ice thicknesses presented in Fig. 6. Given the steep bathymetry suggested by Marchenko et al. (2021), this seems to be a substantial tidal deflection relative to the ice thicknesses.

We agree with the reviewer. The modified manuscript (see above lines 134 -137 and 195-197) now states that cracks are tidal cracks.

Fig. 3c shows calendar day occurrences for icequakes. The authors note that "The icequakes have calendar occurrences every day of the deployment, but are more frequent between February 27th and March 13th, and then between March 21$^{st}$ and March 25$^{th}$" (lines 118–119). Spring tides occurred on Mar 6 and Mar 21, 2019, and a neap tide on Mar 13.

Hourly tidal data from Nylesund, 157 km away, shows tidal heights that are visually well-correlated with the icequake occurrence plots in Fig 3c (http://uhslc.soest.hawaii.edu/data/csv/fast/hourly/h823.csv) The decreased occurrence rates in the latter half of March 2019 may be due to the thickening (and, presumably, strengthening) of the lake ice (Fig. 6b).

We thank reviewer 2 for pointing towards the tides chart at Ny Alesund. With have added this chart to figure 3 (see figure 3-f) to make it comparable with figure 3c. There is a correlation with icequakes occurrences indeed.

The roughly uniform occurrence of icequakes throughout the summed days shown in Fig 3d for cluster family 0–7 could be explained by the hourly precession of high tides throughout the month. Tidal minimums during the Mar 13 neap tide occurred at 0800–0900 UTC, coincident with the 9AM (local? UTC?) decrease in icequakes noted on line 120.

This is a very good point and this is now mentioned in the manuscript:

the other 6 clusters of the first family. The icequakes have calendar occurrences every day of the deployment, but are more frequent between February 27$^{th}$ and March 13$^{th}$, and then between March 21$^{st}$ and March 25$^{th}$ (figure 3c). It is noteworthy that this is consistent with the tides chart shown in figure 3f, and also with the fact that spring tides occurred on March 6 and March 21, while a neap tide occurred on March 13. Icequakes occur at all time of the day with the same temporal distribution, except

130     around 9 AM where occurrences are slightly decreased (figure 3d). The decreased occurrence rates in the latter half of March could be due to the thickening of the ice ($\sim 25\%$ thickness increase).

Lines 150–154 : Forward modeling uses the flexural-gravity wave dispersion relation from Stein et al. (1998). This equation assumes an infinite depth water column. Vallunden Lake has a depth of no more than 10 m (Marchenko et al. 2013). Ewing and Crary (1934) provide a formulation for FG waves in shallow water. Based on a comparison of the two dispersion relations, the Stein formulation diverges from the shallow water case for frequency-thickness products less than 1 Hz m. The current study uses icequakes in the 1–50 Hz range, with a peak at 8 Hz (presumably; the authors do not explicitly state the frequency band for their inversions). Their results are likely not significantly impacted by the assumption of infinite depth. However, the regime change—and their avoidance of it—should nonetheless be acknowledged, especially given that they do acknowledge the high frequency regime change for frequency-thickness > 1000 Hz m.

Reviewer 1 also asked to clarify this assumption. We are copying here the answer made to reviewer 1.

We assume that ignoring the finite water depth of 10 m has a negligible effect on the inverted thickness, based on a comparison between the model used in the manuscript by Stein et al. (1998) and the model by Romeyn et al. (2021). See for example the following figure. Wavenumbers are almost identical in the frequency range of interest.

[Figure]

Wavenumber vs frequency for the QS mode, calculated in a 60 cm-thick ice sheet floating on water. Blue solid line: model by Stein et al. (1998) based on an infinite water depth. Black dashed line: model by Romeyn et al. (2021), based on water with a 10 m depth.

We have added the following sentence, at the beginning of section 3.1 to explain this.

165    1. given a set of parameters for source position around the array (latitude and longitude), source activation time, and ice thickness, generate the synthetic waveforms of the $QS$ mode at the geophones. Synthetic waveforms are generated based on a Ricker wavelet that is propagated in the ice using the analytical, low-frequency asymptotic model by Stein et al. (1998), with the following ice mechanical properties: Young's modulus = 3.8 GPa, Poisson's ratio = 0.28, and density = 910 kg/m$^3$ (Serripierri et al., 2022). This model cannot account for the finite water depth of about 10 m, like the
170    model by Romeyn et al. (2021) can, but by comparing both models, we have checked that this has negligible effect at the frequencies of interest.

Future investigations will be made using a forward model based on the spectral element method, that accounts for finite water depth and for snow cover.

Lines 208–209, 296–298 : The authors state that their method could be adapted for near real time monitoring of ice thickness. What actual time scales are envisioned for a data product? Hourly? Daily?

It is possible that the reviewer did not have the latest version of the manuscript. The latest version mentions: "it could be possible to generate a similar map for each day, hence achieving near real-time maps of sea ice thickness evolution". But this is actually dependent on the number of icequakes recorded every day. We also recorded many icequakes on drifting pack ice (See Moreau et al. 2020b). The text was modified to emphasize this point:

the ice thickness. We demonstrate the possibility of generating maps of sea ice thickness and microseismic activity, with a
50    temporal resolution that is directly linked to the amount of icequakes recorded. In the specific configuration at the fjord, icequakes occurrences are driven by tide. On drifting ice, icequakes are generated by other mechanisms such as swell or ice motion, and many icequakes are also present in the ambiant seismic field (Moreau et al., 2020b). With hundreds of icequakes recorded everyday, a daily temporal resolution can be achieved. We also use the energy information to calculate the scaling law of icequakes in terms of their released energy.

Line 221 : The jumps are stated to be 1 m. How was this guaranteed? Was the jumpee stepping off a 1 m platform? Was the impact onto un-groomed snow & ice, or onto a strike plate? A standing jump does not have sufficient repeatability to be classified as a "calibrated source."

We made standing jumps directly on the ice after removing the snow, hence we cannot guarantee that the height is accurate down to the centimeter, but the goal here is to study the repartition of the orders of magnitude of the icequakes. In this context, an error of a few centimeters would be negligible. We did not realize that the term "calibrated" would be controversial. We have replaced it by "artificial."

Line 260 : The authors mention expanding further work to include longer period ice waves, to the order of 0.1 Hz m. Such waves would absolutely interact with the lake bottom and thus necessitate the Ewing and Crary (1934) or similar formulation.

The frequencies of investigation were adapted to the dimensions of the Lake. This statement is about applying the method to drifting pack ice. We have made the sentence clearer: "Hence, ==on drifting pack ice==, by adjusting the size of the geophones antenna…"

**References**

Cole, Hank M. Tidally Induced Seismicity at the Grounded Margins of the Ross Ice Shelf, Antarctica. Diss. Colorado State University, 2020.

Crary, A. P. "Seismic studies on Fletcher's ice island, Tâ 3." Eos, Transactions American Geophysical Union 35.2 (1954): 293-300.

Ewing, Maurice, and A. P. Crary. "Propagation of elastic waves in ice. Part II." Physics 5.7 (1934): 181-184.

Marchenko, A. V., and E. G. Morozov. "Asymmetric tide in Lake Vallunden (Spitsbergen)." Nonlinear Processes in Geophysics 20.6 (2013): 935-944.

Marchenko, A. V., et al. "Ice thickening caused by freezing of tidal jet." Russian Journal of Earth Sciences 21.2 (2021): 5.

**Technical Comments**

Due to the revisions recommended, this is section is withheld. In general, the grammar and organization are clear and concise.

---

## Referee Report (RR1)

Line 64 : Suggest the more commonly used "flexural gravity wave" which 1) emphasizes the importance of buoyancy, 2) differentiates it from the (strictly elastic) flexural plate wave approximation commonly used in mechanical engineering & ultrasonics, and 3) will improve search results for this paper.

In general, I would argue against using the 'quasi' terminology for a seismology paper since those terms are mostly restricted to ultrasonic NDT references. In this case, the authors do make a valid justification based on consistency with prior works by the same author (which *were* in the ultrasonics regime).

Line 69 : "one-to-one" implies a linear (m=1) relationship. Suggest simply "direct".

Line 121–122 : "another stations". Should be "other stations" or "another station".

Lines 144–146 : Incongruous statement from previous version suggesting that the icequakes are temperature-driven.

Line 155–156 : Please specify UTC vs Local. UTC is mentioned in Line 154, but the use of "noon" on Line 156 makes this ambiguous since noon is generally only meaningful for local time.

Line 169 : Suggest "…we briefly recall the inversion method…"

Line 204 : Orphaned sentence fragment.

Line 210 : "which mass was 39 tons". Suggest "with a mass of 39 tons" or "for which the mass was 39 tons" or "whose mass was 39 tons".

Line 240 : "isolate inversions for which source position…"

Line 243 : "The amplitude of a guided wave…"

Line 269 : "Gutenberg-Richter"

Lines 273 : "…between the icequakes source and the 5 geophones…"

Line 307 : Suggest "…that can telemeter the continuous recordings via satellite…"

Line 317 : "The analysis consists of a two-steps…" ('step' should be singular, at least in American English.)

Line 321 : "…the average ice thickness between the sources and the geophones." Should be plural for consistency of "seismic sources" in first half of sentence.

Line 323–324 : Incongruous statement from previous version suggesting that the icequakes are temperature-driven.

Line 334 : "…waveform inversions strategies…" Inversion should be singular.

---

## Author Response (AR2)

**Response to reviewer 1**

The authors have made a commendable effort revising the submitted manuscript. There are a few remaining issues regarding failures in consistency that have been introduced due to changing interpretation in the revised manuscript and the interpretation of thickness estimates compared with drilled thicknesses that are now provided explicitly. These issues are detailed below:

We thank the reviewer for the second review, and we have made the requested changes to the manuscript. Please see below a detailed answer to each point. We hope that this revision will grant publication to our manuscript.

Line 144-146 "At the location of the deployment, semidiurnal tide reaches 10-20 cm, so it is likely that tides have less effect than changes in temperature between day and night, because the majority of icequakes occurs with a period of 24 hours (figure 6). This periodicity can also be seen in figure 3c, especially between March 1st and March 15th." This should be removed as it doesn't fit with the preceding interpretation. Maybe this was an oversight incorporating the revised interpretation into the original manuscript?

This part of the manuscript was meant to be removed in the revised version indeed. Thank you for pointing this out.

Line 204: "due to thermal expansion and the mechanical tension caused by tide." This also appears to be a sentence fragment from the original manuscript that can now be deleted. Correct, we have removed this sentence

Line 282-286: Judging from Figure 7, it seems that the flexural wave thickness estimates in this study indeed systematically underestimate compared to drilled thickness by 5-10 cm. Perhaps this should be incorporated into the discussion here? The systematic underestimation of the flexural wave estimates is the dominant impression I am left with from Figure 7, and this seems like an essential point to discuss and try to understand. I agree that the relative increase in estimated thicknesses over the study period seems to agree well with the increase in drilled thickness. Another likely reason the drilled thicknesses are larger than the flexural wave estimates is heterogeneous ice, i.e., the drilled thickness likely includes a layer of superimposed snow-ice with much lower elastic modulus than the underlying columnar ice. In this sense, the flexural wave estimates could be considered effective thicknesses for the assumed elastic properties, though consideration of a sandwich plate with varying elastic properties may facilitate a more precise interpretation than the rather vague concept of "effective thickness". Perhaps the authors made some observations in the field that could shed further light on why the drilled thicknesses appear to be larger than the flexural wave estimates?

The heterogeneity of the ice through the thickness was indeed reported in Moreau et al. (2020a), with much porosity at the surface (compacted snow), and much denser ice at depth. We also discussed that this could affect our thickness and Young's modulus estimations, that would actually correspond to "effective" values.

We have, however, explained in the discussion part that using plate theory-based models systematically lead to underestimation of the ice thickness as soon as frequency content above 10 Hz is used in our inversions. The numerical model allows a snow layer to be added on top of the ice. Snow makes the speed of the flexural wave lower and results in underestimation of the ice thickness by a few cm as well. The combined effects of snow and plate theory limitations explain the discrepancies

between ice drillings and estimations, which can also be a consequences of spatial thickness vairations by the way.

This is now discussed further in the revised manuscript.

280     This model, like all models based on plate theory, also suffers limitation of being restricted to low frequencies. Ongoing comparisons between inversions using this model and a full numerical model based on the spectral element method (Cao et al., 2021) suggest that using a model based on plate theory underestimates the ice thickness by a few cm, as soon as the frequency band of interest includes frequencies above 10 Hz, for an ice thickness of 1 m. Moreover, these numerical investigations also reveal that the snow layer, if not accounted for in the model, lead to some estimations that reflect "apparent values" for the

285   ice thickness and mechanical properties (Moreau et al., 2020a). In particular, the snow layer introduces a gradient of porosity through the thickness which makes the flexural wave velocity lower, resulting in potential underestimation of the ice thickness.
    These are, however, preliminary results and the investigation is still ongoing. The full study will be presented in details in a separate paper. These limitations to the model (plate theory and not accounting for snow), together with the spatial hetero-geneity of the ice thickness in the field, explain why the thickness estimations appear to be slightly less that those measured by

290   drilling the ice (figure 7b).

Line 302: Can we expect that the Scatseisnet clustering will give the same arrangement of clusters/families for icequakes and noise sources for polar pack ice in the open Arctic ocean? Given that automation is a key goal of the study, it might be worth adding that manual intervention may still be required during an initial calibration phase at a given field site, in order to interpret the type of events that the automatically extracted clusters correspond to?
Yes, this is expected. Scatseisnet was actually also applied to data recorded during the Damocles expedition on drifting pack ice (2007), and to data recorded on a lake in Finland near Lammi (Lake Pääjäärvi 2021 and 2022). Similar clusters were obtained using identical parameters for each dataset. See the following figure for a screenshot of icequakes waveforms in the clusters obtained from these datasets.

[Figure]

Scatsesinet waveforms for icequakes clusters obtained on drifting pack ice during the DAMOCLES expedition in 2007 (left), and on a lake near Lammi in Finland in 2021 (right).

Obviously, the icequakes cluster from data in 2007 have lower SNR, but it was still possible to obtain thickness estimates (with an average of about 2 m) that were consistent with drillings (see Moreau et al, 2020b).

Different parameters settings would be necessary only when the duration of the events to cluster are of different orders of magnitude (less than a sec, a few seconds, or long-lasting events), which is not the case for icequakes, that all last between 1s and a few seconds.

Line 323-324: "occurs with a recurrence of about 24 hours. This indicates that cracking is likely associated with thermomechanical forcing resulting mainly from both temperatures changes between day and night" Is this another fragment of the original manuscript that was missed upon revision of the interpretation to tidally forced cracking? As written, it doesn't fit with the rest of the manuscript.

This was overlooked after the modifications made to the original manuscript, and is now changed to be consistent with tidal icequakes.

Figure 7: Drilled thicknesses are distinctly along the upper range of the flexural wave estimates. Can this be due to an azimuth effect as Figure 8 indicates can lead to systematic differences in thickness estimates? Or does this relate to the low-frequency effect discussed from line 282-286? It would be useful to annotate the positions of the drill holes on Fig. 7a. See also comment on Line 282-286.

Ice drilling positions are shown in figure 1b of the revised manuscript, and are restricted to the area inside the geophone's locations. It is possible that ice thickness was slightly larger in this area.
We believe that the small discrepancies between the estimations and drillings are mainly associated with modelling limitations. Azimuthal differences in Figure 8 only reflect the natural spatial variations of ice thickness that were also observed in the field.

**Response to reviewer 2**

We have made all the requested corrections, and we thank the reviewer for pointing these out in the second review.

Line 64 : Suggest the more commonly used "flexural gravity wave" which 1) emphasizes the importance of buoyancy, 2) differentiates it from the (strictly elastic) flexural plate wave approximation commonly used in mechanical engineering & ultrasonics, and 3) will improve search results for this paper.

We have made this change.

In general, I would argue against using the 'quasi' terminology for a seismology paper since those terms are mostly restricted to ultrasonic NDT references. In this case, the authors do make a valid justification based on consistency with prior works by the same author (which *were* in the ultrasonics regime).

Yes, this terminology is to remain consistent with our previous papers.

Line 69 : "one-to-one" implies a linear (m=1) relationship. Suggest simply "direct". Line 121–122 : "another stations". Should be "other stations" or "another station".

Lines 144–146 : Incongruous statement from previous version suggesting that the icequakes are temperature- driven.

Line 155–156 : Please specify UTC vs Local. UTC is mentioned in Line 154, but the use of "noon" on Line 156 makes this ambiguous since noon is generally only meaningful for local time.

Line 169 : Suggest "...we briefly recall the inversion method..." Line 204 : Orphaned sentence fragment.

Line 210 : "which mass was 39 tons". Suggest "with a mass of 39 tons" or "for which the mass was 39 tons" or "whose mass was 39 tons".

Line 240 : "isolate inversions for which source position..."

Line 243 : "The amplitude of a guided wave..."

Line 269 : "Gute**n**berg-Richter"

Lines 273 : "...between the icequakes source and the 5 geophones..."

Line 307 : Suggest "...that can telemeter the continuous recordings via satellite..."

Line 317 : "The analysis consists of a two-step**s**..." ('step' should be singular, at least in American English.)

Line 321 : "...the average ice thickness between the source**s** and the geophones." Should be plural for consistency of "seismic sources" in first half of sentence.

All done

Line 323–324 : Incongruous statement from previous version suggesting that the icequakes are temperature- driven.

Line 334 : "…waveform inversions strategies…" Inversion should be singular.